# How Does Tropospheric VOC Chemistry Affect Climate? An Investigation of Preindustrial Control Simulations Using the Community Earth System Model Version 2

Noah A. Stanton[1] and Neil F. Tandon[1]

[1]Department of Earth and Space Science and Engineering, York University, Toronto, Ontario, Canada

**Correspondence:** Noah A. Stanton (nstant@my.yorku.ca)

**Abstract.** Because of their computational expense, models with comprehensive tropospheric chemistry have typically been run with prescribed sea surface temperatures (SSTs), which greatly limits the model's ability to generate climate responses to atmospheric forcings. In the past few years, however, several fully-coupled models with comprehensive tropospheric chemistry have been developed. For example, the Community Earth System Model version 2 with the Whole Atmosphere Community Climate Model version 6 as its atmospheric component (CESM2-WACCM6) has implemented fully interactive tropospheric chemistry with 231 chemical species as well as a fully coupled ocean. Earlier versions of this model used a "SOAG scheme" that prescribes bulk emission of a single gas-phase precursor to secondary organic aerosols (SOAs). In contrast, CESM2-WACCM6 simulates the chemistry of a comprehensive range of volatile organic compounds (VOCs) responsible for tropospheric aerosol formation. Such a model offers an opportunity to examine the full climate effects of comprehensive tropospheric chemistry. To examine these effects, 211-year preindustrial control simulations were performed using the following two configurations: 1) the standard CESM2-WACCM6 configuration with interactive chemistry over the whole atmosphere (WACtl), and 2) a simplified CESM2-WACCM6 configuration using a SOAG scheme in the troposphere and interactive chemistry in the middle atmosphere (MACtl). The middle atmospheric chemistry is the same in both configurations, and only the tropospheric chemistry differs. Differences between WACtl and MACtl were analyzed for various fields. Regional differences in annual mean surface temperature range from -4 K to 4 K. In the zonal average, there is widespread tropospheric cooling in the extratropics. Longwave forcers are shown to be unlikely drivers of this cooling, and possible shortwave forcers are explored. Evidence is presented that the climate response is primarily due to increased sulfate aerosols in the extratropical stratosphere and cloud feedbacks. As found in earlier studies, enhanced internal mixing with SOAs in WACtl causes widespread reductions of black carbon (BC) and primary organic matter (POM), which are not directly influenced by VOC chemistry. These BC and POM reductions might further contribute to cooling in the Northern Hemisphere. The extratropical tropospheric cooling results in dynamical changes, such as equatorward shifts of the midlatitude jets, which in turn drive extratropical changes in clouds and precipitation. In the tropical upper troposphere, cloud-driven increases in shortwave heating appear to weaken and expand the Hadley circulation, which in turn drives changes in tropical and subtropical precipitation. Some of the climate responses are quantitatively large enough in some regions to motivate future investigations of VOC chemistry's possible influences on anthropogenic climate change.

# 1    Introduction

The Intergovernmental Panel on Climate Change Sixth Assessment Report (IPCC-AR6) assessed contributions to observed warming since industrialization. This assessment found that the direct, indirect, and semi-direct effects of aerosols are altogether the largest sources of uncertainty for anthropogenic radiative forcing (IPCC, 2021). Secondary organic aerosols (SOAs) account for a significant portion of aerosols in the troposphere. SOAs are in turn produced through a range of chemical reactions involving volatile organic compounds (VOCs), and the spatial distributions of SOAs and VOCs are coupled to atmospheric dynamics. Thus, understanding the climate effects of tropospheric VOC chemistry is critical to reducing uncertainty in assessing contributors to both past and projected climate change.

VOCs include biogenic VOCs (BVOCs) emitted by plants (e.g. isoprene, monoterpenes, sesquiterpenes) and anthropogenic VOCs, and there are estimated to be 10,000 to nearly 1 million different organic compounds in the atmosphere (Mahilang et al., 2021; Guenther et al., 2012). VOCs are oxidized in the atmosphere by several different chemical species, such as hydroxyl, nitrate, chlorine radicals and ozone (Finlayson-Pitts and Pitts Jr, 1999). The oxidation of VOCs in the atmosphere leads to the production of SOAs. Presently, biogenic SOAs make up the majority of SOA loading ($\sim$88 $\mathrm{TgC\,y^{-1}}$) (Srivastava et al., 2022). The oxidation reactions are typically initiated at a double-bond in the VOC, creating peroxy radicals (Ziemann and Atkinson, 2012). These peroxy radicals ($RO_2$) can participate in further oxidation reactions, producing a range of products, such as organic nitrates ($RONO_2$) and hydroperoxyl radicals ($HO_2$) (Srivastava et al., 2022; Schwantes et al., 2015). Given this complex web of chemical interactions, computational models are crucial for understanding the climate effects of tropospheric chemistry.

While 3-D chemical transport models (CTMs), like GEOS-Chem, do exist and provide comprehensive organic chemistry, the chemistry by design is decoupled from the atmospheric dynamics: the transport model is usually run offline with prescribed wind fields, and chemical constituent output is then prescribed in an Earth system model (ESM) or general circulation model (GCM) (Murray et al., 2021). Extremely complex atmospheric oxidation chemistry models also exist, like the Generator for Explicit Chemistry and Kinetics of Organics in the Atmosphere (GECKO-A), and they provide detailed explicit chemistry of organic species and multi-generational chemistry (Lannuque et al., 2018). However, GECKO-A is typically run as a box model due to its computational expense.

In the past few years, however, several models have been developed that allow for comprehensive treatments of both tropospheric chemistry and other aspects of the Earth system while maintaining full coupling between the model components (e.g., Yukimoto et al., 2019; Dunne et al., 2020; Sellar et al., 2020; Wu et al., 2020; Miller et al., 2021). One such model is the Community Earth System Model version 2 with the Whole Atmosphere Community Climate Model version 6 as its atmospheric component (CESM2-WACCM6). Previous versions of this model included a more limited representation of organic chemistry compared to that in CESM2-WACCM6 (Danabasoglu et al., 2020). Climate simulations with more comprehensive chemistry typically prescribe sea surface temperatures (SSTs) (e.g., Tilmes et al., 2019), which facilitates calculating standard climate change metrics like effective radiative forcing (ERF) (e.g., Forster et al., 2016), but this approach limits the ability of the model to simulate climate responses to atmospheric forcings. CESM2-WACCM6, however, is capable of including both

a dynamical ocean as well as comprehensive VOC chemistry in the troposphere. This model simulates interactive chemistry between organics and other atmospheric constituents, including aerosols (sulfate aerosol, salt, dust, etc.). These improvements to tropospheric chemistry also have an impact on gas-phase products such as $O_3$ and NOx ($NO + NO_2$).

These recently-developed modelling capabilities present a new opportunity to investigate the climate effects of tropospheric VOC chemistry in a fully coupled model framework. In this study, we take advantage of this opportunity by comparing a control simulation of the standard CESM2-WACCM6 with a control simulation using a simplified CESM2-WACCM6 configuration in which detailed VOC chemistry is not included and emissions of a single bulk gas-phase SOA precursor are prescribed (referred to as a "SOAG scheme"). The difference between these two simulations provides an indication of the climate effect of tropospheric VOC chemistry. Past studies have compared the climates produced by CESM2 configurations with and without comprehensive chemistry in the troposphere (Gettelman et al., 2019; Danabasoglu et al., 2020). However, those CESM2 configurations also differed in their stratospheric chemistry, the number of model levels, the altitude of the model top as well as the treatment of the quasi-biennial oscillation (QBO). This study is distinct in that it more precisely isolates the effect of tropospheric VOC chemistry without other confounding factors.

Tilmes et al. (2019) have compared configurations of CESM2 that also precisely isolate the effect of tropospheric VOCs, but their simulations used prescribed SSTs and sea ice, which as mentioned before, limits the model's ability to produce a full climate response. Nonetheless, the findings of Tilmes et al. (2019) are very helpful for interpreting results of our simulations, as we will discuss below. Most of the simulations performed under the Aerosol Chemistry Model Intercomparison Project (AerChemMIP) also used prescribed SSTs (Collins et al., 2017). Among these prescribed SST simulations, there may be simulations that allow for assessment of the effects of VOC chemistry in other models (albeit without simulating the full climate response to VOC chemistry), and we plan to investigate those simulations further in the future. The AerChemMIP simulations that did not prescribe SSTs included experiments in which aerosols were prescribed and interactive. Comparison of such simulations, while also of great interest, would not isolate the effect of VOC chemistry, which is our focus here. Aerosols in are prognostic in all of our simulations, but our model configurations differ in the chemical details with which aerosols are produced.

This study complements the extensive research focusing on the effects of stratospheric chemistry and particularly stratospheric ozone (e.g., Son et al., 2009; Polvani et al., 2011; Previdi and Polvani, 2014; Calvo et al., 2015). When the effects of tropospheric VOC chemistry are isolated, our simulations show significant impacts on climate, comparable in some regions to the temperature changes expected from anthropogenic greenhouse gas (GHG) increases over a century in future climate change scenarios. Even though only the tropospheric chemistry differs between our simulations, we will show that there are significant impacts over both the troposphere and stratosphere, and there are significant impacts on species that are not directly influenced by VOC chemistry, such as black carbon (BC) and primary organic matter (POM). In particular, we find that, compared to the SOAG scheme, the explicit VOC chemistry in CESM2-WACCM6 results in sulfate aerosol increases throughout the stratosphere and generally cooler temperatures in the extratropics. We will present evidence suggesting that these sulfate changes are primarily influenced by hydroxyl and SOA changes.

Despite the comprehensive chemistry in CESM2-WACCM6, there are (as detailed below) important limitations in this aspect as well as other aspects of the model (such as convective parameterization). So we cannot be certain that the climate effects highlighted in this study represent the full effects of tropospheric VOC chemistry in the real world. Rather, we consider this study to be a first step in disentangling the possible climate effects of VOC chemistry, which can help highlight areas for future model development and observational studies.

We begin by describing CESM2-WACCM6 and our experimental setup (Section 2). Then we show the response to VOC chemistry, first focusing on the spatial structure of changes in key climate parameters (Section 3), followed by investigation of the zonal mean response (Section 4). The possible constituents responsible for the climate response are discussed in Section 5, along with the possible mechanisms responsible for changes in those constituents. Further details regarding cloud effects are examined in Section 6, and Section 7 provides a summary and concluding remarks.

## 2 Methods

### 2.1 The Community Earth System Model version 2 (CESM2)

CESM2 was used for this study because, in addition to being a fully-coupled climate model with comprehensive atmospheric chemistry, it is open source and has extensive documentation and support. CESM2 consists of 7 components: land, river runoff, surface waves, ocean, land ice, sea ice and atmosphere. All of these components are coupled to each other through the Common Infrastructure for Modelling the Earth version 5 (CIME5; Danabasoglu et al., 2020).

The ocean component of CESM2 is the Parallel Ocean Program version 2 (POP2), which is a level-coordinate ocean general circulation model (Danabasoglu et al., 2012). Surface waves are represented with the National Oceanic and Atmospheric Administration's WaveWatch-III (WW3) ocean surface wave prediction model (Tolman et al., 2009). The sea ice component is the Los Alamos sea ice model (CICE), which includes both thermodynamic and dynamical sea ice processes (Hunke et al., 2015). The land ice component is the Community Ice Sheet Model (CISM), which is a 3-D thermomechanical model that calculates momentum balance, thickness and temperature of ice sheets (Lipscomb et al., 2019). The land component is the Community Land Model version 5 (CLM5), which includes ecological and land use processes as well as a fire module that generates emissions from biomass burning (Lawrence et al., 2019). River runoff is calculated using the Model for Scale Adaptive River Transport (MOSART). MOSART uses a kinetic wave method that is derived from the mass and momentum equations (Li et al., 2013).

The atmospheric component of CESM2 can be configured as the Community Atmospheric Model version 6 (CAM6) or WACCM6 (Danabasoglu et al., 2020), which includes more comprehensive chemistry compared to CAM6. In this study, WACCM6 was chosen as the atmospheric component.

**Table 1.** CESM2-WACCM6 configurations used in this study.

| Case Name | Compset | Atmospheric Chemistry | SOA precursors | QBO Treatment |
|-----------|---------|----------------------|----------------|---------------|
| WACtl | BW1850 | whole atmosphere | interactive VOCs | nudged |
| MACtl | BWma1850 | middle atmosphere | SOAG scheme | nudged* |
| MACtl-CMIP6 | BWma1850 | middle atmosphere | SOAG scheme | interactive |

*This differs from the default configuration under the BWma1850 compset, which includes a freely evolving QBO.

## 2.2   The Whole Atmosphere Community Climate Model version 6 (WACCM6)

WACCM6 captures a large number of atmospheric chemical reactions from Earth's surface up to the lower thermosphere.
There are 70 model levels, using a hybrid sigma vertical coordinate system, with a model top at an atmospheric pressure of
$4.5 \times 10^{-6}$ hPa (around 130 km). WACCM6 has 231 prognostic chemical species with numerous chemical pathways for the
whole atmosphere, including detailed modelling of SOAs. WACCM6 also uses the Modal Aerosol Model (MAM4), which is
able to capture complex aerosol-cloud interactions (Danabasoglu et al., 2020). GHGs, ozone, aerosols and clouds are coupled
to a modified version of the Rapid Radiative Transfer Model (RRTM) (Clough et al., 2005) called the Rapid Radiative Transfer
Model for GCMs (RRTMG). In simulations that include anthropogenic emissions, chlorofluorocarbons (CFCs) are also coupled
to RRTMG, but CFCs are not present in our preindustrial simulations.

CESM2-WACCM6 was run in two different configurations, as summarized in Table 1. The "WACtl" case has comprehensive chemistry over the whole atmosphere, including VOC chemistry in the troposphere. In contrast, the "MACtl" case has
comprehensive chemistry in the middle atmosphere and simplified chemistry (without explicit non-methane VOC reactions)
in the troposphere. The $CH_4$ oxidation reactions are the same in WACtl and MACtl, with the only difference being that a
reaction that produces $CH_4$ from $O_3$ and propene is not present in MACtl. Hereafter, we will use the term "VOCs" to refer to
non-methane VOCs. The simplified SOA scheme in MACtl is known as a SOAG scheme, and it is commonly implemented
in GCMs. Under this scheme, anthropogenic and biogenic precursor VOCs are assumed to have fixed mass yields and are
sorted into five primary VOC bins. Within MAM4, these gas-phase precursors are lumped into a single semi-volatile organic
gas-phase species called SOAG, which then condenses into primary aerosol particles (Liu et al., 2012; Emmons et al., 2010).
One point is worth emphasizing here: it would be misleading to say that VOCs are entirely absent in MACtl. Rather, explicit
VOC chemistry is absent in MACtl, but MACtl still captures the effects of VOCs themselves through the SOAG scheme. Thus,
the difference between WACtl and MACtl relates to explicit VOC chemistry, not to the presence of VOCs themselves.

Standard configurations of CESM2 are given labels called "compsets" that aid reproducibility by defining relevant details
of the specific model configuration, including physics, microphysics, chemistry and input datasets. The WACtl case uses
the BW1850 compset (which is the standard configuration for CESM2-WACCM6), and the MACtl case uses the BWma1850
compset. The resolution specification for both MACtl and WACtl is f19_g17, meaning that the atmosphere and land components
are on a $1.9° \times 2.5°$ horizontal grid, while the ocean and sea ice are on a displaced Greenland pole grid with approximately

1° resolution. By default, the BWma1850 compset has a freely evolving QBO, while the BW1850 compset has a QBO that is nudged to observed winds. We set the QBO winds to be nudged in both configurations to avoid confounding effects on our results due to QBO differences. The archive for the Coupled Model Intercomparison Project phase 6 (CMIP6; Eyring et al., 2016) includes a 500-year preindustrial control simulation using the unmodified BWma1850 compset with interactive QBO (variant label r1i1p1f1). We refer to this simulation as "MACtl-CMIP6," and we will include some analysis of this simulation below.

The BW1850 compset implements the troposphere, stratosphere, mesosphere, and lower thermosphere (TSMLT) chemistry set. The BWma1850 compset uses the middle atmosphere, mesosphere, and lower troposphere (MAMLT) chemistry set. The chemistry set used in MAMLT is similar to the chemistry set used in WACCM4 (with 81 solution species), with the additions of two metastable states of $O^+$ (Marsh et al., 2013; Gettelman et al., 2019). As mentioned before, this compset does not include any explicit non-methane VOC chemistry in the troposphere, and it instead prescribes SOAG emissions. The BW1850 compset implements 231 solution species (including tropospheric VOCs) with 583 chemical reactions broken into 150 photolysis reactions and 403 gas-phase reactions. In this compset, peroxyacetyl nitrate (PAN) is considered part of organic nitrate in the total nitrogen budget (Emmons et al., 2020). Organic nitrates that condense are lumped with sulfate aerosol. Even though the stratospheric chemistry is the same in both compsets, the additional tropospheric chemistry in BW1850 has an impact on stratospheric composition, as we will examine further below.

CESM2-WACCM6 also simulates a number of heterogeneous reactions in the troposphere and stratosphere. MAM4 includes tropospheric heterogeneous reactions involving four aerosol types: sulfate, BC, POM and SOA. The four aerosol types are grouped together to generate a single surface area density (SAD) quantity that is used in these heterogeneous reactions. Four tropospheric heterogeneous reactions are included in both compsets, and BW1850 includes an additional nine heterogeneous reactions involving VOCs. For the latter reactions, aerosols do not directly participate in VOC chemistry, but aerosols and VOCs can interact indirectly through changes in SAD. Both compsets include 17 stratospheric heterogeneous reactions, which occur with three aerosol types: sulfate, nitric acid trihydrate (NAT) and water-ice. Tropospheric oxidants ($O_3$, $OH$, $NO_3$ and $HO_2$) are explicitly calculated in both compsets (Gettelman et al., 2019). Both compsets include wet and dry deposition of BC, POM, sea salt, sulfate, dust and SOA. When POM and BC are emitted in the primary carbon mode, the condensation of $H_2SO_4$ and semi-volatile organics results in POM and BC being transferred into the accumulation mode, which is an irreversible process. Additionally, these species are transferred from the primary carbon mode into the accumulation mode by coagulation with the Aitken and accumulation modes (Liu et al., 2012).

The intermediate semivolatile organic compounds produced by these chemical reactions are grouped into volatility bins collectively referred to as a volatility basis set (VBS) that parameterizes the SOA formation process. Hodzic et al. (2016) describe the modified 1-D VBS scheme implemented in WACCM6. The older SOA model used in MACtl is a simpler VOC oxidation parameterization with four oxidation bins for oxygenated semi-volatile organic compounds and two additional bins for SOA aged from anthropogenic precursors and gas-phase oxidized VOCs (OVOCs). A rate constant is prescribed for the chemical aging of anthropogenic oxidation intermediates by OH. The biogenic precursors in the model are not artificially aged. The VOC chemistry in WACtl requires a more detailed treatment of SOA formation, which in turn requires a more

complex VBS scheme. The updated VBS used in WACtl derives its SOA formation mechanism (the oxidation curve) from the Statistical Oxidation Model (SOM) first described by Cappa and Wilson (2012). This SOM implements improved SOA yields that account for wall-losses in chamber studies. SOM prognoses the multi-generational chemistry of SOAs, with fragmentation and functionalization included. The reader is referred to (Tilmes et al., 2019) for a detailed comparison of available aerosol and BC observations against CESM2-WACCM6 configurations with these different VBS schemes.

One notable chemical species that is missing from both WACtl and MACtl is nitrous acid (HONO). HONO can photolyze via the reaction $HONO + hv \rightarrow OH + NO$ ($\lambda < 400$ nm), and it contributes to the formation of hydroxyl radicals. HONO reactions are particularly relevant in polluted urban areas where concentrations can be up to a few parts per billion. Primary anthropogenic sources of HONO are combustion processes, while primary natural emissions come from soil microbial activity and biocrusts. Secondary sources of HONO (chemically produced in the atmosphere and surface), are from gas-phase homogeneous reactions of NO and OH during the daytime or heterogeneous reactions with $NO_2$ at the surface (Kramer et al., 2020). The exclusion of HONO, especially in simulations of future climate, may impact the budget of OH and NO.

## 2.3 External forcings

Both MACtl and WACtl are control runs with external forcings fixed to "preindustrial" levels, taken to correspond to 1850 C.E. conditions. This is a standard baseline period used in climate assessments such as those performed by the IPCC (e.g., IPCC, 2021). Thus, our simulations do not include most anthropogenic emissions since industrialization. In this study, we focus on the effects of VOC chemistry on this preindustrial climate state because it is likely simpler to understand, as it does not include the time-dependent forcings of anthropogenic climate change. Furthermore, examining VOC chemistry effects on a preindustrial climate provides initial hypotheses and points of comparison to facilitate insights in studies that examine the effects of VOC chemistry on anthropogenic climate change.

In our experiments, surface emissions of anthropogenic GHGs, reactive gases (including NOx from non-lightning sources) and aerosols, as well as volcanic emissions ($SO_2$ and sulfate aerosol) come from emissions specified under CMIP6 (Gettelman et al., 2019; Meinshausen et al., 2017). These emissions include the portions due to anthropogenic biomass burning that are not interactively calculated by the fire module within CLM5 (van Marle et al., 2017). In all cases, these external forcings are prescribed as a repeating annual cycle (corresponding to year 1850) with monthly time resolution. In MACtl, the prescribed SOAG is derived from specified CMIP6 surface emission data for five primary VOCs using the following fixed mass yields: 25% monoterpenes, 15% aromatics, 5% BIGALK (lumped butanes and larger alkanes), 5% BIGENE (lumped butenes and larger alkenes) and 4% isoprene (Liu et al., 2012; Tilmes et al., 2019). In contrast, WACtl calculates biogenic VOC surface emissions using the Model of Emissions of Gases and Aerosols from Nature (MEGAN) version 2.1 (Guenther et al., 2012). In both WACtl and MACtl, NOx emissions from lightning are interactively generated (about 3-4 $TgN\,y^{-1}$).

## 2.4 Simulation execution and output analysis

Each case was run for 250 y, and data from years 40-250 inclusive (spanning 211 y) were used in our analysis. The first 39 y of output were discarded because they showed significant trends in global mean surface temperature, indicating that the

model had not fully equilibrated. The two cases were not initialized from the same ocean and sea ice states, and in the future, we plan to examine possible sensitivity to different initialization approaches. The MACtl case was initialized from another preindustrial control simulation using the BWma1850 compset. In the WACtl case, the ocean and sea ice were initialized from a 1955-2012 climatology derived from the World Ocean Atlas 2013 version 2 (Locarnini et al., 2013). Neither of our simulations produces unrealistic Labrador sea ice cover, as found with particular initializations of this model (Danabasoglu et al., 2020). The difference between the WACtl and MACtl initial temperature states is similar to a difference between a La Niña state and an El Niño state, and the initial sea ice state differences follow these temperature differences (not shown). Thus, the difference between the WACtl and MACtl initial states appears to be a difference in the phase of internal variability, rather than a fundamental difference in climate state, and we do not expect that we would obtain significantly different long-term averages if our simulations were initialized from exactly the same state.

We assess the effects of VOC chemistry on climate by computing the difference between the annual average of WACtl and the annual average of MACtl. Hereafter, this difference will be referred to as the "mean difference" between WACtl and MACtl, or simply "WACtl – MACtl." For some fields, we compute percent changes, in which case this difference is divided by the climatological value in the MACtl simulation. The mean differences were further analyzed to determine if those differences were statistically significant. The statistical significance test involved two steps: First, the number of the independent samples in each dataset was identified, and then a two-tailed t-test was performed on these samples. For each timeseries at each location, independent samples were identified using the lag-1 autocorrelation technique described by Bretherton et al. (1999).

All plots in the zonally averaged pressure-latitude plane show the MACtl climatology of the tropopause pressure level. This quantity was calculated by taking a zonal mean of the MACtl thermal tropopause pressure (WACCM6 output variable "TROP_P"). The difference between the zonal mean MACtl tropopause and the zonal mean WACtl tropopause is small (at most 3 hPa) and is not displayed in the figures.

## 3  Spatial structure of the VOC chemistry effect

What are the effects of tropospheric VOC chemistry on basic climate characteristics, such as surface temperature, clouds, precipitation and winds? Figure 1 depicts the mean difference (WACtl – MACtl) in surface temperature, with reds indicating warming and blues indicating cooling. Between New Zealand and South America, there is 0.5-1.5 K warming with 1-4 K cooling over the rest of the Southern Ocean. There is warming ranging from 0.5 to 2 K from Siberia across the Arctic Ocean through to the Barents and Greenland Seas, with weaker warming over central Asia and central North America. Surface temperatures cool 0.5-1.2 K off the coast of Japan, with accompanying warming over the Gulf of Alaska. There is more localized strong warming (3-4 K) in southeast Asia, the southwestern US and the Great Lakes region. Localized pockets of cooling intersperse with pockets of warming over central and southern Africa, eastern Australia and central South America. The warming in particular regions is comparable to that generated by anthropogenic GHG increases over a century in future climate change scenarios (e.g., Lu et al., 2008), which motivates further research to examine possible effects of VOC chemistry on anthropogenic climate change. While significant changes in surface temperature are seen over land, the localized mechanisms

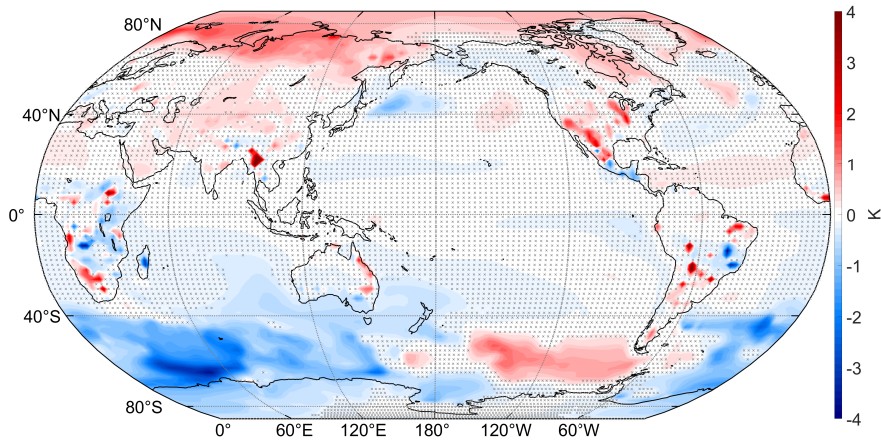

**Figure 1.** Annual mean difference (WACtl – MACtl) in surface temperature (K). Gray crosses indicate responses that are statistically insignificant, with uncrossed responses being statistically significant at the 95% level.

behind these changes are not clear and will be investigated in future studies. Overall, the surface temperature response shows no resemblance to the La Niña–like difference in the initial states of WACtl and MACtl. (See discussion in section 2.4.) This finding further establishes that the WACtl – MACtl responses are not being influenced by differences in their initial states.

The mean difference in global mean surface temperature (GMST) between WACtl and MACtl is $-0.0640$ K over the 211 y analysis period. This result indicates that, while VOC chemistry has a slight global cooling effect, there are much larger regional temperature effects that strongly cancel with each other when averaging spatially. When imposing a climate change forcing, it is common to calculate the ERF to assess the strength of the forcing compared to forcings from other sources. A common method for calculating the ERF is the regression method described by Gregory et al. (2004). However, there is an initial warming of GMST in the first $\sim$20 years of WACtl relative to MACtl (for reasons that are unclear), even though the steady-state GMST response is negative, suggesting that the transient and equilibrium radiative forcings are opposite in sign. In such a situation, the regression method is not suitable for calculating the ERF, and simulations with fixed SSTs should instead be used (e.g., Hansen et al., 2005; Forster et al., 2016). Such simulations along with further investigation of the contrasting transient and equilibrium responses will be left for future work.

Taking a difference between our WACtl simulation and a 211 year segment of the MACtl-CMIP6 simulation (which, as detailed in section 2.2, includes an interactive QBO) produces nearly identical results to our Fig. 1 (not shown), which suggests that the treatment of the QBO has negligible effect on the surface temperature response. Given that the climates of MACtl and MACtl-CMIP6 are essentially the same, the 500 years of MACtl-CMIP6 output also allows for additional assessment of the noise associated with internal climate variability. We estimate this noise by averaging the surface temperature over a 211-year slice of MACtl-CMIP6 and then subtracting the surface temperature averaged over a non-overlapping 211-year slice of the same simulation. Fig. 2 shows these time-slice differences for four different choices of time range, obtained by shifting

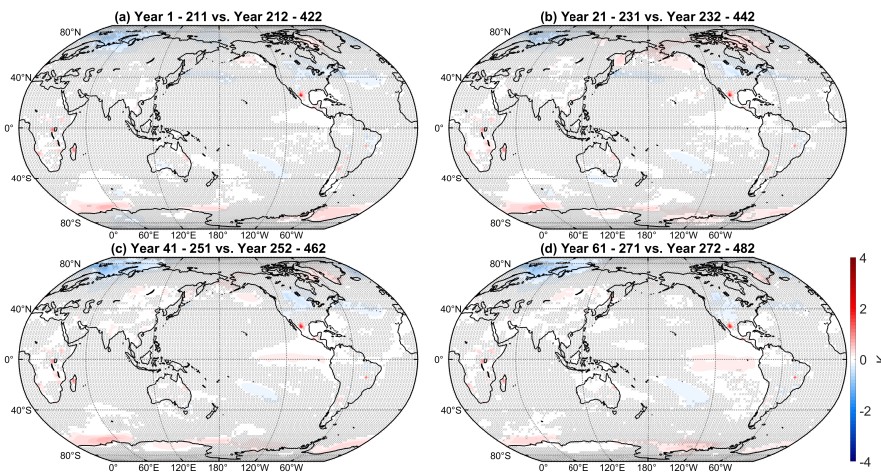

**Figure 2.** Difference of time-averaged surface temperature between non-overlapping 211-year time slices of the MACtl-CMIP6 simulation (see text for additional details). The precise simulation years that were averaged are indicated in the title over each panel. Gray crosses indicate responses that are statistically insignificant, with uncrossed responses being statistically significant at the 95% level.

the beginning of each time slice by 20 years, with the results plotted on the same shading scale as in Fig. 1. The resulting differences for all of the time slice choices are negligible compared to the significant temperature differences shown in Fig. 1. In addition to our statistical significance tests, this analysis further demonstrates the robustness of our results.

A key influence on regional climate is the large-scale atmospheric circulation, which can be partially depicted through sea level pressure (SLP). Figure 3b shows the MACtl climatology of sea level pressure along with the approximate location of the midlatitude jets (in green), defined as the latitude where there is a local maximum in surface wind speed. In accordance with geostrophic balance, the midlatitude jets form on the boundaries between the polar lows and subtropical highs. Figure 3a shows the mean difference in SLP, with blues indicating decreases and reds indicating increases. Over the South Pacific between New Zealand and South America, there is a weakening of the subtropical high accompanying an overall weakening of the polar low. These changes indicate a weakening of the meridional pressure gradient, which we expect to result in a weakening and equatorward shift of the midlatitude jet. (This shift will be confirmed further below when examining zonal mean zonal wind.) There is some strengthening of the subtropical high over the Indian Ocean, but this SLP change is smaller than the weakening of the polar low, and the meridional pressure gradient still decreases. Throughout the Northern Hemisphere (NH), the subtropical highs and polar lows weaken, also suggesting a weakening and equatorward shift of the midlatitude jet.

Regional precipitation changes can be due to changes in thermodynamics as well as dynamics. Figure 4 shows the mean difference of daily precipitation, with blues indicating increases and reds indicating decreases. There are especially strong changes in precipitation around the Intertropical Convergence Zone (ITCZ): Precipitation increases up to $1.5 \, \mathrm{mm \, d^{-1}}$ over the northern and eastern portions of the Maritime Continent, with accompanying decreases ($0.5$-$1 \, \mathrm{mm \, d^{-1}}$) over the central and eastern equatorial Pacific. This band of decreased precipitation extends eastward to central Africa.

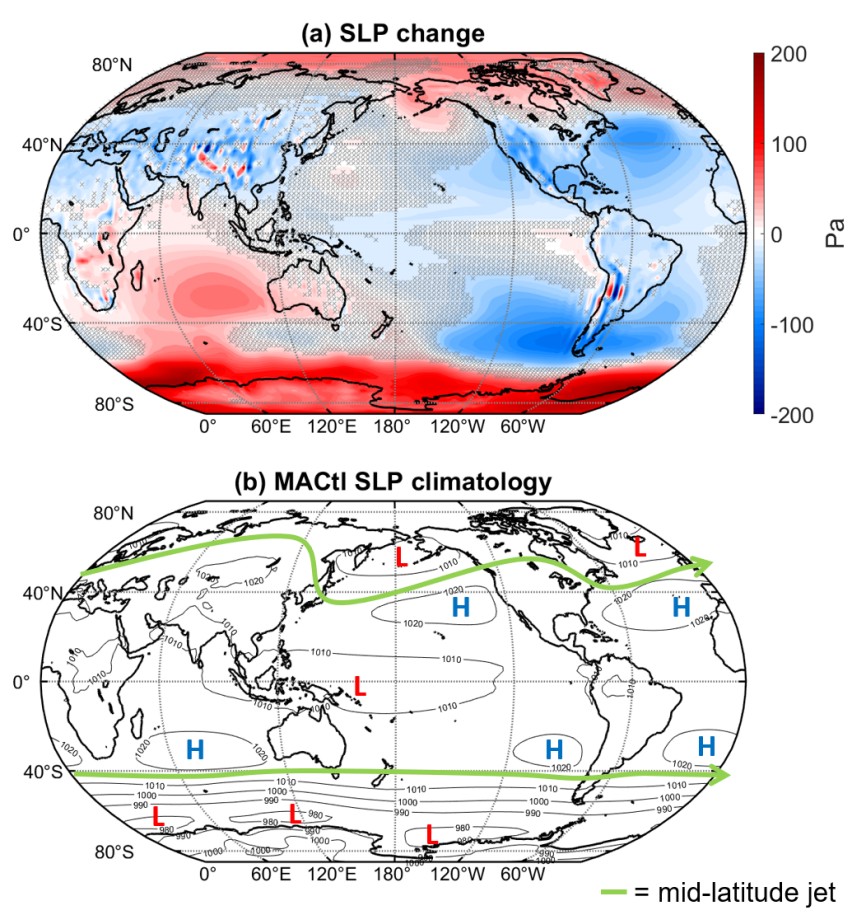

**Figure 3.** (a) Sea level pressure (Pa) mean difference (WACtl – MACtl). Gray crosses indicate responses that are statistically insignificant, with uncrossed responses being statistically significant at the 95% level. (b) MACtl sea level pressure climatology with lows labelled as L and highs labelled as H. The green line marks the approximate location of the midlatitude jets, where there is a local maximum in surface wind speed.

The precipitation changes over the equatorial Pacific are what would be expected with a strengthening of the Walker circu-
lation (e.g., Dong and Lu, 2013). Such a strengthening of the Walker circulation occurs under La Niña, along with cooling in the eastern equatorial Pacific (e.g., Power and Smith, 2007). However, Fig. 1 does not show significant cooling in the eastern equatorial Pacific, which suggests that other processes are responsible for the strengthened Walker circulation, a matter that requires further investigation.

Accompanying the precipitation decreases along the equator, there is a band of increased precipitation just north of the
equator extending from the Atlantic eastward to Eritrea, indicating a northward shift of the ITCZ. Over the Americas, the equatorial precipitation decrease is accompanied by precipitation increases to the north and south, indicating a widening of the

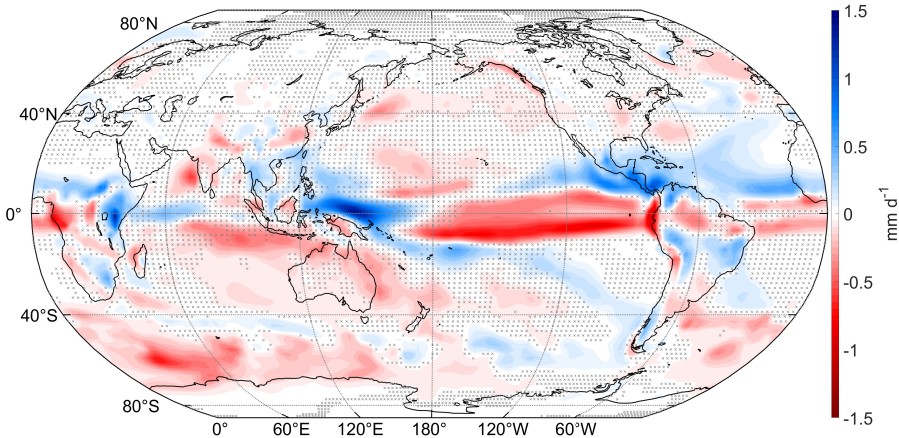

**Figure 4.** Annual mean difference (WACtl – MACtl) in daily precipitation ($\mathrm{mm\,d^{-1}}$). Gray crosses indicate responses that are statistically insignificant, with uncrossed responses being statistically significant at the 95% level.

ITCZ. Over the Indian Ocean, precipitation increases over the equator with accompanying decreases to the north and south, indicating a narrowing of the ITCZ. ITCZ shifts have been shown to relate to changes in the global energy transport (e.g., Kang et al., 2008). Given the apparent regional variability in the ITCZ changes in our simulations, there are likely region-specific changes in energy transport at work (cf. Mamalakis et al., 2021), which require further investigation.

There are precipitation decreases over most of the Southern Ocean (up to $0.7\ \mathrm{mm\,d^{-1}}$). These changes align with SLP increases over the Southern Ocean (Fig. 3a), with which we expect anomalous descent and decreased precipitation. Precipitation decreases over the Indian subcontinent, which corresponds with a weakening of the north-south pressure gradient in this region (Fig. 3a), indicating weaker flow from the Arabian Sea toward the Himalayas. Off the east coast of Japan, there are precipitation decreases of 0.1 to 0.4 $\mathrm{mm\,d^{-1}}$ which align with surface temperature cooling in that region (Fig. 1), suggesting a primarily thermodynamic effect.

Over the subtropical South Pacific, roughly from Samoa to southern Chile, there is an arc of increased precipitation (up to $0.25\ \mathrm{mm\,d^{-1}}$) with an accompanying decrease to the south and west. There are qualitatively similar dipole changes in precipitation over the subtropics in the North Pacific and Atlantic. These subtropical precipitation changes do not align with significant temperature anomalies, indicating that they are likely driven by dynamical changes, such as changes in the Hadley circulation. Such changes cannot be related simply to SLP changes, and we will investigate the relevant dynamical changes further below.

Changes in clouds have a significant influence on the surface energy budget and hence temperature. Figure 5a shows the mean difference in total cloud fraction, and Figure 5b shows the mean difference in net cloud radiative effect (net CRE), defined as the difference between the all-sky and clear-sky net radiation at the top of the atmosphere. Blues indicate decreases, while reds indicate increases. Over most of the extratropics, cloud fraction changes and CRE changes are opposite in sign,

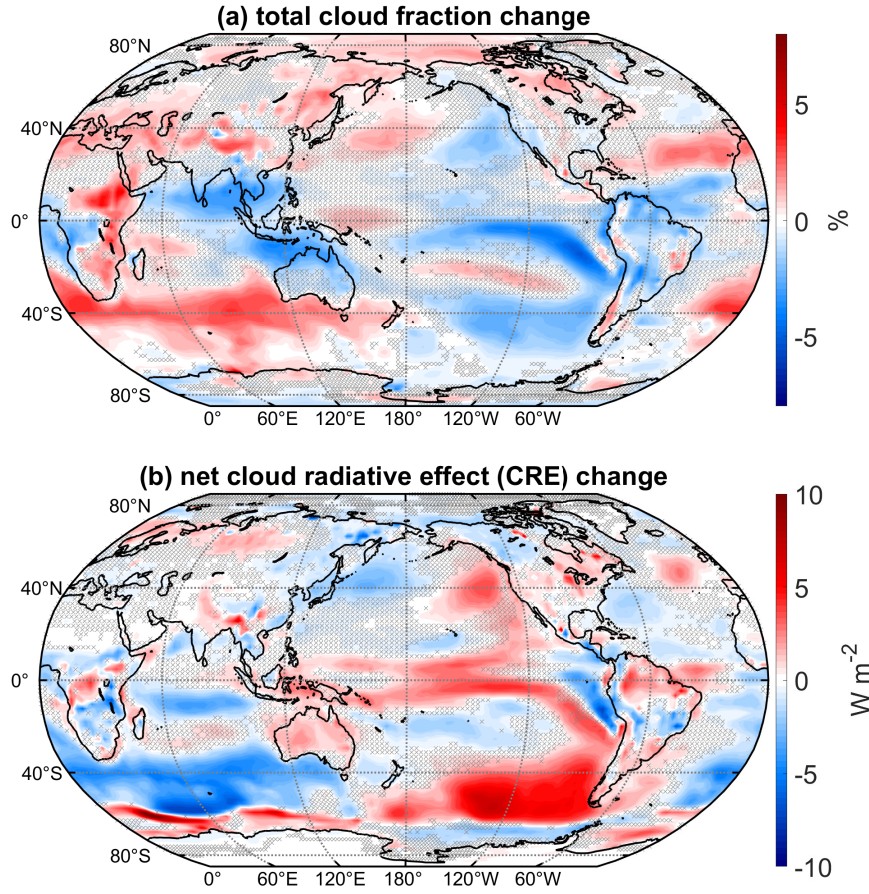

**Figure 5.** (a) Total cloud fraction mean difference (WACtl – MACtl). (b) Net cloud radiative effect mean differences (WACtl – MACtl). Gray crosses indicate responses that are statistically insignificant, with uncrossed responses being statistically significant at the 95% level.

as is expected in regions where low clouds predominate. In regions where there is large-scale subsidence, such as the eastern equatorial Pacific and subtropical dry zones, clouds are also confined to lower altitudes, and cloud fraction and CRE changes are opposite in sign.

In tropical regions of deep convection, however, such as the western equatorial Pacific and the equatorial Indian Ocean, there is not so clear a correspondence between cloud fraction and CRE changes, as there are competing effects of low and high cloud changes. As was found previously when examining precipitation changes (Fig. 4), the cloud fraction changes over the equatorial Pacific show evidence of a strengthened Walker circulation, with cloud fraction increasing in the west and decreasing in the east. The dipole cloud fraction changes in the subtropics suggest a possible influence of changes in the Hadley circulation, which will be considered further below.

Over the Southern Ocean between New Zealand and South America, there are moderate (up to 3%) decreases in the total cloud fraction with 2 to 7 W m$^{-2}$ increases in CRE. This CRE increase shows a spatial correspondence with warming over this region (Fig. 1), suggesting that this warming is due to positive cloud feedbacks. The deceleration and equatorward shift of the midlatitude jet would be expected to reduce cloud cover in this region, and we will discuss this possibility further below.

Elsewhere over the Southern Ocean, there are moderate increases (up to 4%) in the total cloud fraction with CRE decreases of up to 8 W m$^{-2}$. Intriguingly, SLP increases throughout this region (Fig. 3), with which one might expect anomalous descent and reduced cloud cover. This result suggests that the cloud increases in this region are not dynamically driven and are instead a thermodynamic result of cooler temperatures resulting in a downward shift of the clouds. We will explore this possibility further below. As well, the surface temperature cooling in this region extends beyond the regions of CRE decrease. For example, 335 there is cooling between Madagascar and Australia (Fig. 1), despite a CRE increase in this region. This provides evidence that the cooling over the broader region encompassing the eastern Southern Ocean is not primarily driven by cloud feedbacks, although cloud feedbacks appear to amplify the cooling here. Furthermore, if atmospheric dynamics were driving the temperature changes here, one would expect either a dipole surface temperature anomaly or warming associated with anomalous adiabatic descent, neither of which is evident in Fig. 1. Overall, these results suggest that the cooling over this region is primarily due to 340 non-cloud radiative effects rather than dynamical effects.

South of 60°S and from 150°E eastward to 60°W, there are decreases in CRE, and between 60°W eastward to 150°E there are increases in CRE. These changes show correspondence with changes in sea ice concentration (SIC; Fig. A1), which changes in accordance with the surface temperature changes (Fig. 1). That is, the CRE decreases (increases) here closely correspond with SIC decreases (increases). This makes sense because, if low cloud cover is fixed (which is approximately true around 345 Antarctica) and SIC decreases (increases), then the surface albedo decreases (increases), and the radiative effect of low clouds gets stronger (weaker), implying a negative (positive) CRE change. The correspondence between CRE and SIC changes is less apparent around the date line, and consideration of other factors such as sea ice albedo and thickness changes might be required here, a topic for future work.

Over most of the NH extratropics, there are increases in cloud fraction (up to 3%) with corresponding CRE decreases of up 350 to 3.5 W m$^{-2}$. However, surface temperatures warm over most of this area (Fig. 1), suggesting that clouds are not driving these temperature changes. The warming over the northern extratropical oceans shows some correspondence with SLP increases (Fig. 3), suggesting that adiabatic warming is playing a role, and we would expect the warming of these bodies of water to generate warming of the surrounding land regions. In particular, the decreased SLP over Eurasia combined with the increased SLP over the pole would favour advection of warm anomalies from the Arctic Ocean toward Eurasia. We also cannot rule out 355 that non-cloud radiative processes are contributing to the warming in the Northern extratropics, and further investigation of the detailed causes is required.

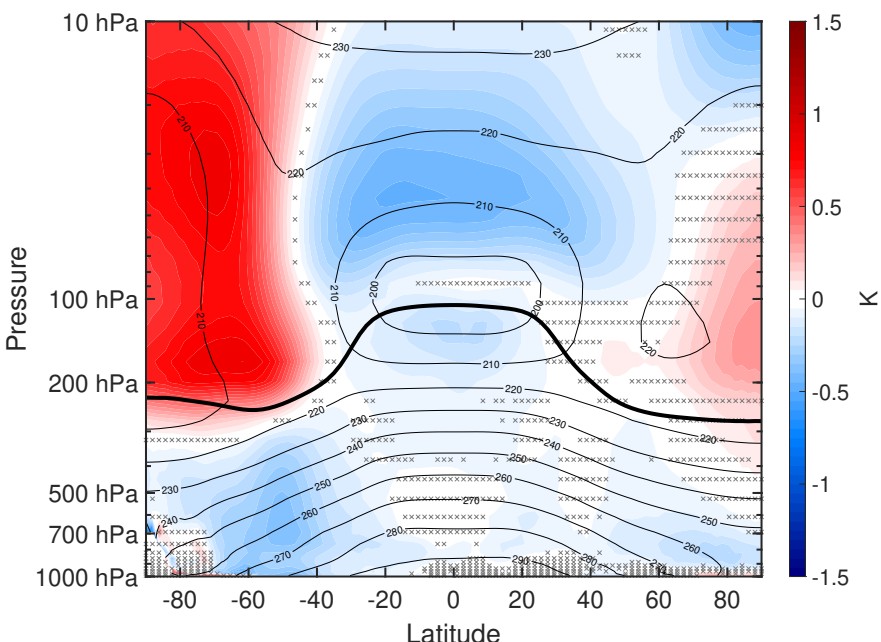

**Figure 6.** Annual mean difference (WACtl – MACtl) in zonal mean atmospheric temperature (K). The black contours indicate the MACtl climatology of air temperature (K), with a contour interval of 10 K. Gray crosses indicate responses that are statistically insignificant, with uncrossed responses being statistically significant at the 95% level. The thick black line marks the MACtl climatology of the tropopause.

## 4  Zonal-mean structure of VOC chemistry effect

The previous figures provided a global picture of the impacts of tropospheric VOC chemistry, and we can gain additional insight by examining the zonally-averaged vertical structure of changes in key parameters. Figure 6 shows the mean difference
of zonal mean atmospheric temperature, with reds indicating increases and blues indicating decreases. There is 0.25-0.5 K cooling throughout the tropical stratosphere and Southern Hemisphere (SH) midlatitude troposphere. There is weaker cooling over most of the NH troposphere, which contrasts with surface temperature warming over most of NH (Fig. 1). Less apparent in Fig. 6 is that there is 0.05-0.31 K warming in the lowest three model levels north of 60°N. So at these latitudes, there is a transition from warming in the boundary layer to cooling in the free troposphere.

As noted previously, changes in large scale dynamics play a key role in shaping the climate response to VOC chemistry. Figure 7a gives further detail of the dynamical changes by showing the mean difference of zonal wind, with reds indicating increases and blues indicating decreases. In SH, the stratospheric polar vortex weakens, and this negative zonal wind anomaly extends down into the troposphere on the poleward flank of the midlatitude jet. There is an accompanying acceleration on the equatorward flank of the SH midlatitude jet, implying an equatorward shift of the midlatitude jet, further confirming the shift
suggested by the SLP changes examined previously (Fig. 3). These changes follow what is expected from thermal wind balance:

The zonal wind anomalies correspond with warming in the SH high latitudes which, in combination with midlatitude cooling, implies a reduction in the meridional temperature gradient. In NH, there is a qualitatively similar weakening of the stratospheric polar vortex and zonal wind deceleration on the poleward flank of the midlatitude jet. However, the zonal wind response is quantitatively weaker in NH than in SH, and there is no clear indication of an equatorward shift of the NH midlatitude jet. This weaker zonal wind response in NH compared to SH corresponds with the weaker tropospheric cooling and polar stratospheric warming in NH.

Care is needed when comparing our zonal wind responses to those in earlier studies, since the atmospheric circulation response is highly sensitive to the regional structure of temperature changes. For example, idealized simulations have shown that a narrow band of warming around the equator produces equatorward shifts of the midlatitude jets, whereas warming spread over all latitudes produces poleward shifts (Tandon et al., 2013). In our simulations, the zonal wind responses in both hemispheres qualitatively resemble the circulation responses to high latitude warming found in earlier studies (e.g., Son et al., 2008; Butler et al., 2010), and we will explore the mechanisms generating this warming further below. Moreover, these responses are qualitatively opposite to the poleward midlatitude jet shifts produced in simulations of global warming (with warming over all latitudes), including simulations in which SST is uniformly increased (e.g., Chen et al., 2013). Based on such past work, we would expect that, aside from any regional temperature changes, widespread tropospheric cooling would also shift the midlatitude jets equatorward. Indeed, simulations of the response to increased natural aerosols have shown widespread tropospheric cooling and equatorward shifts of the midlatitude jets (e.g., Allen and Sherwood, 2011). Such an effect might be contributing in NH, where the tropospheric cooling is more spatially uniform than in SH.

Interestingly, the zonal wind increase in the SH midlatitudes lies between the subtropical and midlatitude jets, which in combination with the weaker zonal wind acceleration in the deep tropics, indicates that SH subtropical jet is shifting poleward, opposite to the direction of the midlatitude jet shift. As the subtropical jet approximately corresponds with the edge of the Hadley circulation, this result suggests that the Hadley circulation is changing in ways that contrast with typical expectations based on the midlatitude circulation changes. This result motivates further examination of the mean meridional circulation (MMC), shown in Fig. 7b, with reds indicating positive (clockwise) anomalies and blues indicating negative (counterclockwise) anomalies. The MMC is quantified using the meridional mass streamfunction, calculated by vertically integrating the zonal mean meridional wind (Hartmann, 2015, section 6.3.2). This figure confirms that there is slight expansion of the Hadley circulation in both hemispheres: there is a counterclockwise anomaly extending poleward of the SH HC edge and a clockwise anomaly extending poleward of the NH HC edge.

We have further confirmed these circulation changes using the metrics described in Adam et al. (2018). Specifically, the HC poleward edges in both hemispheres were calculated by finding the zero crossing of the mass streamfunction at 500 hPa. This metric indicates a slight expansion of the HC in both hemispheres ($0.0916°$ in NH, $0.0483°$ in SH), though these changes are not statistically significant. As noted previously when examining Fig. 3, the subtropical high strengthens over the Indian Ocean and weakens elsewhere, which suggests that the HC changes over the Indian Ocean might contrast with those at other longitudes. Indeed, when the longitudes corresponding to the Indian Ocean ($20-150°E$) are excluded from the mass streamfunction calculation, there is stronger, statistically significant HC expansion of $0.5990°$ in SH (not shown). Furthermore,

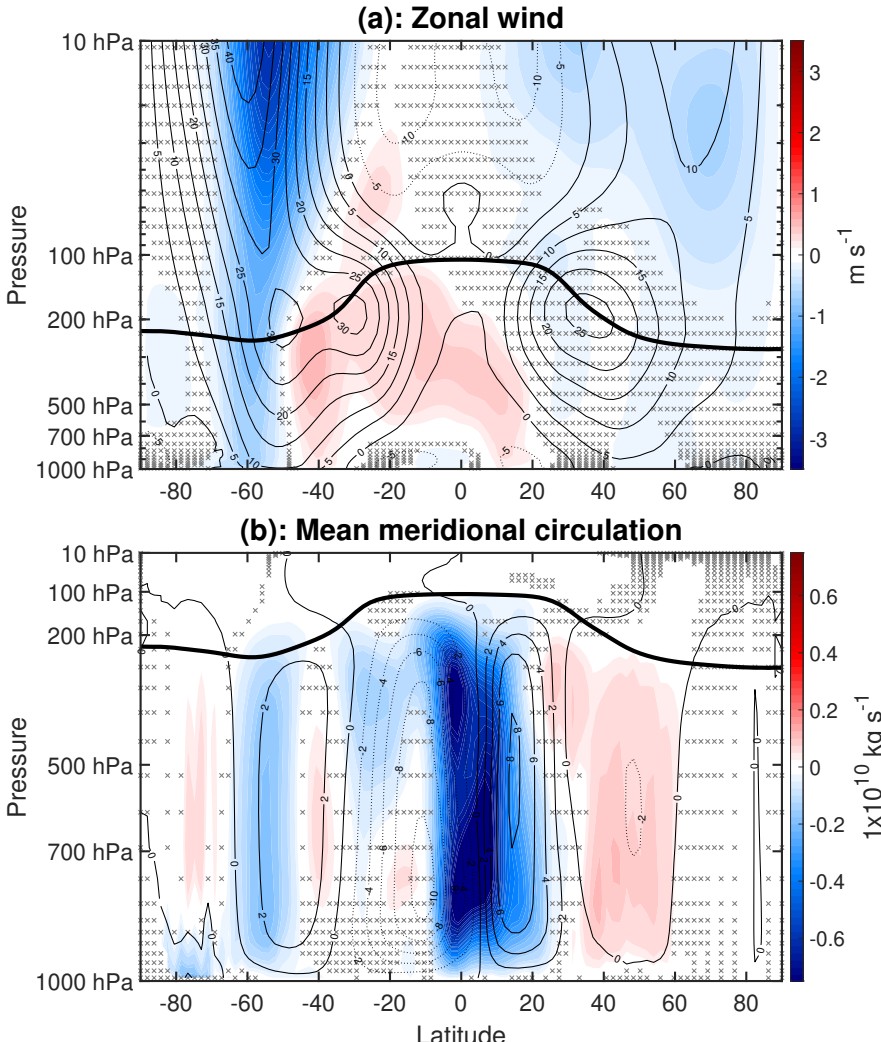

**Figure 7.** (a) Annual mean difference (WACtl – MACtl) in zonal wind ($\mathrm{m\,s^{-1}}$). The black contours indicate the MACtl climatology of zonal wind ($\mathrm{m\,s^{-1}}$), contour interval $5\ \mathrm{m\,s^{-1}}$. (b) Annual mean difference (WACtl – MACtl) in the mean meridional circulation (MMC) ($1\times10^{10}$ $\mathrm{kg\,s^{-1}}$), quantified using the mass streamfunction. The black contours indicate the MACtl climatology of MMC ($1\times10^{10}\ \mathrm{kg\,s^{-1}}$), contour interval $2\times10^{10}\ \mathrm{kg\,s^{-1}}$. Negative values of the MACtl climatology imply counterclockwise circulation and positive values imply clockwise circulation. Gray crosses indicate responses that are statistically insignificant, with uncrossed responses being statistically significant at the 95% level. The thick black line marks the MACtl climatology of the tropopause. Panel (a) uses a logarithmic pressure scale to reveal more detail in the stratosphere while panel (b) uses a linear pressure scale.

the midlatitude jet latitude (the local maximum of zonal wind at the lowest model level) shifts equatorward $0.3868°$ in NH and $0.6231°$ in SH, although the NH shift is not statistically significant. The SH jet shift is about half that produced over a century

in climate change simulations (e.g. Son et al., 2009), motivating further research to investigate the possible effects of VOC chemistry on the large-scale circulation response to anthropogenic climate change.

Fig. 7b also shows that the HC weakens in both hemispheres, although the weakening in NH is much stronger than in SH. In general, weakening of a particular overturning cell is indicated by an anomaly over the central portion of an overturning cell that is opposite in sign to the climatological streamfunction. In this case, HC weakening in NH is indicated by a negative (counterclockwise) anomaly that opposes the positive (clockwise) HC climatology, and HC weakening in SH is indicated by a positive (clockwise) anomaly that opposes the negative (counterclockwise) HC climatology. The counterclockwise anomaly

over the NH HC crosses the equator, indicating a northward shift of the rising branch of the HC, which aligns with the widespread precipitation decreases over the equator and precipitation increases north of the equator (Fig. 4). This weakening and expansion of the HCs might help to explain some of the subtropical precipitation changes in Fig. 4. In particular, one would expect that, with HC expansion, there would be a precipitation decrease poleward of the original HC edge, and the HC weakening would lead to a precipitation increase in the subtropical dry zones, both of which are evident in the subtropical

Atlantic and Pacific. In addition to these HC changes, Fig. 7b also shows weakening of the Ferrel cells in both hemispheres and weakening of the polar cell in SH. As mentioned above, this weakening can be inferred from anomalies that are opposite in sign to the climatology.

To what extent are the temperature changes discussed above radiatively driven versus dynamically driven? Figure 8 shows the mean difference in heating rates due to various sources, with reds indicating increases and blues indicating decreases. The

total radiative heating (Fig. 8a) is calculated by summing the longwave heating (Fig. 8c) and shortwave heating (Fig. 8e). In a steady state, we expect the changes in total radiative heating to be balanced by the changes in dynamical plus latent heating (Fig. 8b). (Here, dynamical heating was computed using WACCM6 variable "DTCORE," which includes both horizontal and vertical temperature advection.) Comparing Figs. 8a and b, this steady-state balance is indeed satisfied. The climatological values of longwave heating are negative (except near the tropical tropopause), which means that positive changes (shaded red)

typically indicate reductions in longwave cooling and negative changes (shaded blue) typically indicate increases in longwave cooling. Comparing Figs. 8a, c and e, we see that most of the changes in total radiative heating are quantitatively accounted for by changes in longwave cooling.

In the stratosphere, the longwave cooling is mostly determined by local changes in blackbody emission, which follows Stefan-Boltzmann's law. Comparing Figs. 8c and 6 in the stratosphere, we accordingly see that regions of increased (de-

creased) longwave cooling correspond with regions of increased (decreased) temperature. Near the surface, the longwave cooling is determined mostly by the difference between the downward longwave flux and the blackbody emission, with the latter dominating. Thus, over most of the lower troposphere, we also see reductions in longwave cooling corresponding with cooler temperature. There are parts of the extratropical mid-troposphere where, despite the reduced blackbody emission due to lower temperatures, the longwave cooling still increases. This change likely results from decreased longwave absorption due

to the reduced upwelling longwave from the lower troposphere.

Comparing Figs. 8c and e, we see that most of the reduced longwave heating in the tropical stratosphere is due to decreased shortwave heating. Elsewhere in the stratosphere, as well as in the tropical upper troposphere and Arctic troposphere, the

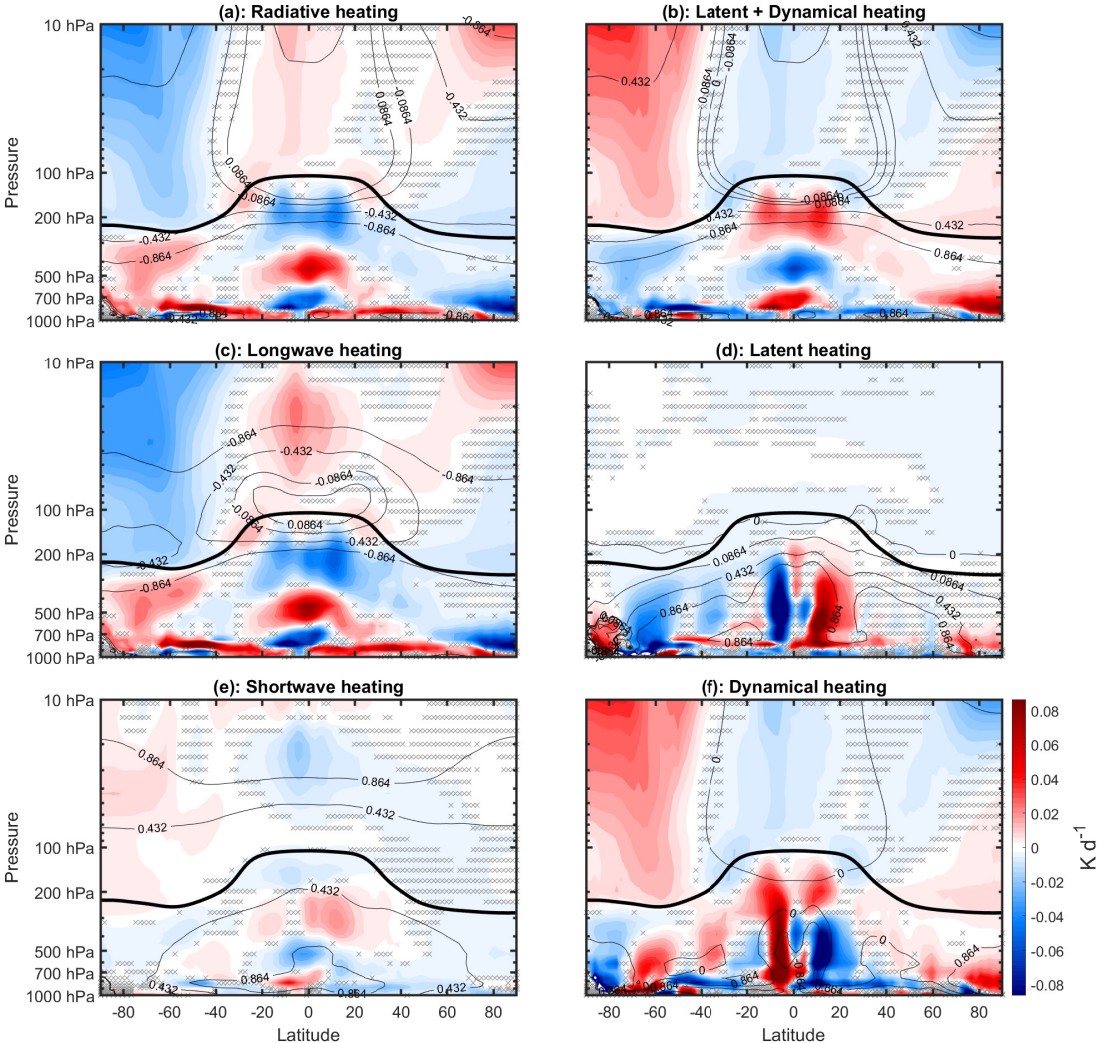

**Figure 8.** Annual mean difference (WACtl – MACtl) in heating rates ($K\,d^{-1}$). Gray crosses indicate responses that are statistically insignificant, with uncrossed responses being statistically significant at the 95% level. The thick black line marks the MACtl climatology of the tropopause. The black contours indicate the MACtl climatology. (a) Total radiative heating, with climatology contours $-0.864$, $-0.432$, $0.432$, $0.864$ $K\,d^{-1}$. (b) Latent plus dynamical heating, with climatology contours $-0.432$, $-0.0864$, $0$, $0.0864$, $0.432$, $0.864$ $K\,d^{-1}$. (c) Longwave heating, with climatology contours $-0.864$, $-0.432$, $-0.0864$, $0.0864$ $K\,d^{-1}$. (d) Latent heating, with climatology contours $0$, $0.0864$, $0.432$, $0.864$ $K\,d^{-1}$. (e) Shortwave heating, with climatology contours $0.432$, $0.864$ $K\,d^{-1}$. (f) Dynamical heating, with climatology contours $0$, $0.864$ $K\,d^{-1}$.

changes in longwave cooling are mostly explained by changes in dynamical heating (Fig. 8f). The anomalous dynamical heating is especially strong in the southern polar stratosphere (SPS), and it dominates over the increased shortwave heating. The increased shortwave heating here appears to be due to increased ozone concentration (Fig. A2). In percentage terms, the largest ozone increases are in the troposphere, especially the ∼25% increases in the tropical and SH upper troposphere. But in absolute terms, the largest increases ($\sim 2.5 \times 10^{11}$ molecules cm$^{-3}$) are in the SPS. The reasons for these ozone changes in the stratosphere and troposphere will be investigated in a separate study. These results can be compared with earlier modelling studies which show that, when stratospheric ozone is increased on its own, the warming in the SPS is dominated by increased shortwave heating (e.g., Keeble et al., 2014) rather than dynamical heating. Altogether, these results suggest that the strong SPS warming in our simulations is dynamically rather than radiatively driven.

Lower in the troposphere, changes in latent heating (Fig. 8d) also have to be considered. Over most of the low and middle troposphere outside of the Arctic, changes in latent heating are qualitatively opposite to changes in dynamical heating. This correspondence suggests that dynamical heating over most of the troposphere is due to vertical motions, as anomalous upward (downward) motion in the troposphere would be expected to produce anomalous adiabatic cooling (warming) as well as increased (reduced) condensational heating. The cancellation, however, is not perfect, and when latent and dynamical heating are summed together (Fig. 8b), it is clear that, over most of the troposphere, the changes in longwave cooling are mostly explained by changes in the sum of latent and dynamical heating. Hereafter, we refer to this combination of dynamical and latent heating as "dynamical-condensational" heating.

As noted above, there is warming near the surface in the polar regions, in contrast with the cooling over most of the extratropical troposphere. Over the Arctic, there is increased longwave cooling around 700 hPa, which we would expect to result in increased downwelling longwave at the surface. Fig. 8 suggests that this increased longwave cooling aloft is due to increased dynamical heating, which would be expected with the anomalous descent resulting from the weakening polar low noted above (Fig. 3). (Shortwave heating decreases in this region, so shortwave fluxes do not appear to be a dominant factor here.) As for the warming near Antarctica, longwave cooling decreases in this region despite the increase in blackbody emission, which suggests that reduced longwave cooling in the upper troposphere is dominating the longwave balance. However, such a change on its own would reduce surface temperature, which suggests that longwave changes do not explain the warming here. Instead, this warming is likely explained by shortwave heating, which does increase here. (This increased shortwave heating is more clearly visible in Fig. 9, which plots the shortwave heating on a linear pressure scale with an adjusted shading scale.) As we will show below, there is not a significant change in CRE here, so the shortwave heating changes near Antarctica are likely due to non-cloud shortwave forcers.

The results above show a strong role for dynamical-condensational changes in driving the longwave cooling changes and temperature response. However, given that the temperature response must ultimately be due to changes in VOC chemistry, there must ultimately be a change in radiative heating that kicks off the dynamical changes. The very close correspondence between the longwave and dynamical-condensational heating changes suggests that the longwave changes are likely dynamically driven. The complex structure of the longwave heating changes further argues for their dynamical origin: Longwave radiative forcers are typically well-mixed and would be expected to produce more spatially uniform longwave cooling changes (e.g., Cai and

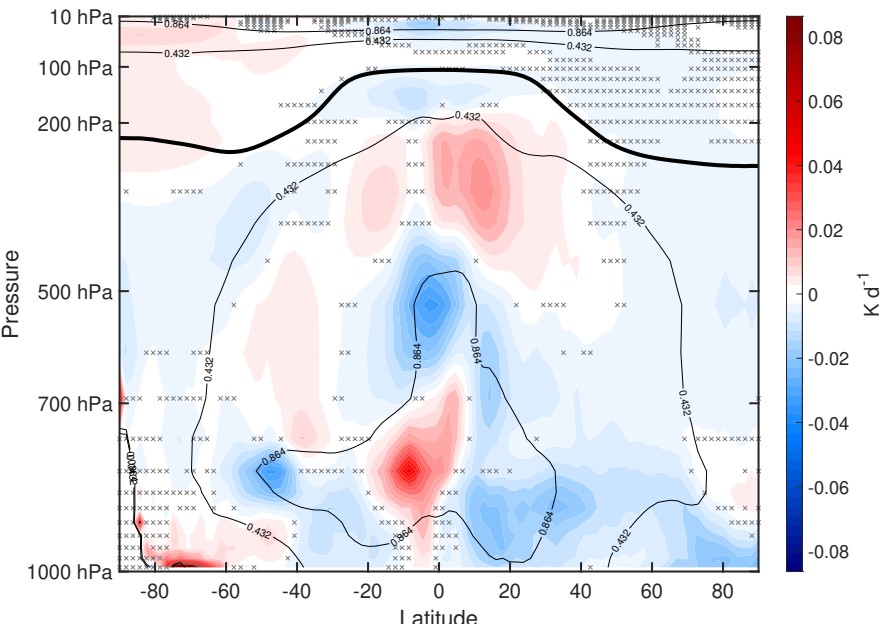

**Figure 9.** Annual mean difference (WACtl – MACtl) in shortwave heating ($K\,d^{-1}$). The black contours indicate the MACtl climatology of shortwave heating, with contours 0.432 and 0.864 $K\,d^{-1}$. Gray crosses indicate responses that are statistically insignificant, with uncrossed responses being statistically significant at the 95% level. The thick black line marks the MACtl climatology of the tropopause.

Tung, 2012, see their Figs.1 and 6). We have also examined changes in methane and carbon dioxide (Fig. A3), and the changes are minuscule in percentage terms and of the wrong sign to explain the temperature responses in our simulations.

The above considerations suggest that the ultimate drivers of the response to VOC chemistry are shortwave forcers, which is expected since VOC chemistry primarily influences tropospheric aerosol formation. As noted above, earlier studies suggest that shortwave forcers are likely not driving the dynamical response in the extratropical stratosphere. However, in the tropical stratosphere, dynamical heating changes are small, and reduced shortwave heating is the dominant driver of the reduced longwave cooling. Thus, one possibility is that a change in a shortwave forcer in the tropical stratosphere drives the tropical

stratospheric cooling which then drives the dynamical response in the troposphere. Indeed, Tandon et al. (2011) have shown that when cooling is imposed in the tropical stratosphere, it results in equatorward shifts of the midlatitude jets, in agreement with our results. However, such a forcing also produces HC contraction and weakening, which is opposite to the responses we obtain in our simulations. This contrast suggests that changes in shortwave heating in the troposphere are likely also contributing to the dynamical changes.

Such a possibility is surprising since the shortwave heating changes are much smaller than the dynamical-condensational changes over most of the atmosphere. Indeed, we previously used this fact to argue that shortwave heating changes were likely not driving the dynamical heating changes in the SPS. But given the stronger effects of dynamical motions, especially

vertical motions, in the troposphere compared to the stratosphere, we cannot rule out that a change in shortwave heating in the troposphere might ultimately produce dynamical-condensational heating changes that are quantitatively larger than the shortwave heating change. The idealized simulations of Cai and Tung (2012) provide support for such a possibility: in the upper troposphere, the warming due to the direct effect of an increase in the solar constant is about a factor of four smaller than the warming due to the changes in convective heating. (Compare their Figs. 3 and 8.) Thus, it is worth taking a closer look at the shortwave heating changes in the troposphere.

Figure 9 shows decreased shortwave heating over most of the extratropical troposphere. Such reductions in shortwave heating could indeed generate cooling in the extratropical troposphere and result in dynamical changes (e.g. equatorward shifts of the jets, weakening of the meridional pressure gradient) that reinforce that cooling. Furthermore, there are more complex changes in radiative heating in the tropics that might explain some of the complexity in the circulation response (i.e. the opposite directions of the midlatitude and subtropical jet shifts).

## 5   Possible shortwave forcers of the climate response

What chemical species are responsible for these changes in shortwave heating? Addressing this question with certainty requires performing controlled experiments, which we plan to do in the future. For the current study, we have examined the patterns of changes in various constituents in order to document key changes and generate compelling hypotheses that can be tested with controlled experiments in the future. After examining numerous possible shortwave forcers, we found that changes in sulfate aerosol, BC and POM are likely contributing to the widespread extratropical tropospheric cooling in our simulations.

VOC chemistry dramatically impacts total SOA concentration. Pure SOAs typically scatter incoming radiation, and increases in SOA burden represent a negative radiative forcing in future climate scenarios (Zhu et al., 2017). Observational studies suggest that SOAs can also be absorbing aerosols (AAs) if internally mixed with BC (Zhang et al., 2018), and such mixing effects are included in CESM2-WACCM6 (Hodzic et al., 2016). Indirect effects of SOA are also highly uncertain, with one study estimating a $0.02 \pm 0.04 \mathrm{~W} \mathrm{~m}^{-2}$ historical radiative forcing due to increased cirrus cloud formation when using an earlier version of CESM with a different aerosol scheme (Zhu and Penner, 2020). As mentioned above, CESM2-WACCM6 includes both direct and indirect radiative effects of SOAs, but the indirect effect in this version of the model has, to our knowledge, not yet been quantified. Furthermore, as mentioned above, SOAs and other aerosols in CESM2-WACCM6 participate in tropospheric heterogeneous reactions that can affect other aspects of atmospheric composition.

Figure 10a shows the mean difference in SOA concentration, with blues indicating decreases and reds indicating increases. In the tropical upper troposphere, there is a 260% increase in SOA concentration. Tilmes et al. (2019) also noted increased SOA in this region when comparing fixed SST simulations of CESM2-WACCM6 with and without comprehensive VOC chemistry. This similarity suggests that the WACtl – MACtl changes in SOA do not ultimately arise from surface temperature changes. Rather, Tilmes et al. (2019) attributed this difference to the faster SOA formation in the SOAG scheme, which results in greater deposition in the lower troposphere and lower SOA concentration in the tropical upper troposphere compared to simulations with explicit VOC chemistry. There is also a ∼20% SOA increase poleward of 50°N in the lower troposphere. Elsewhere in

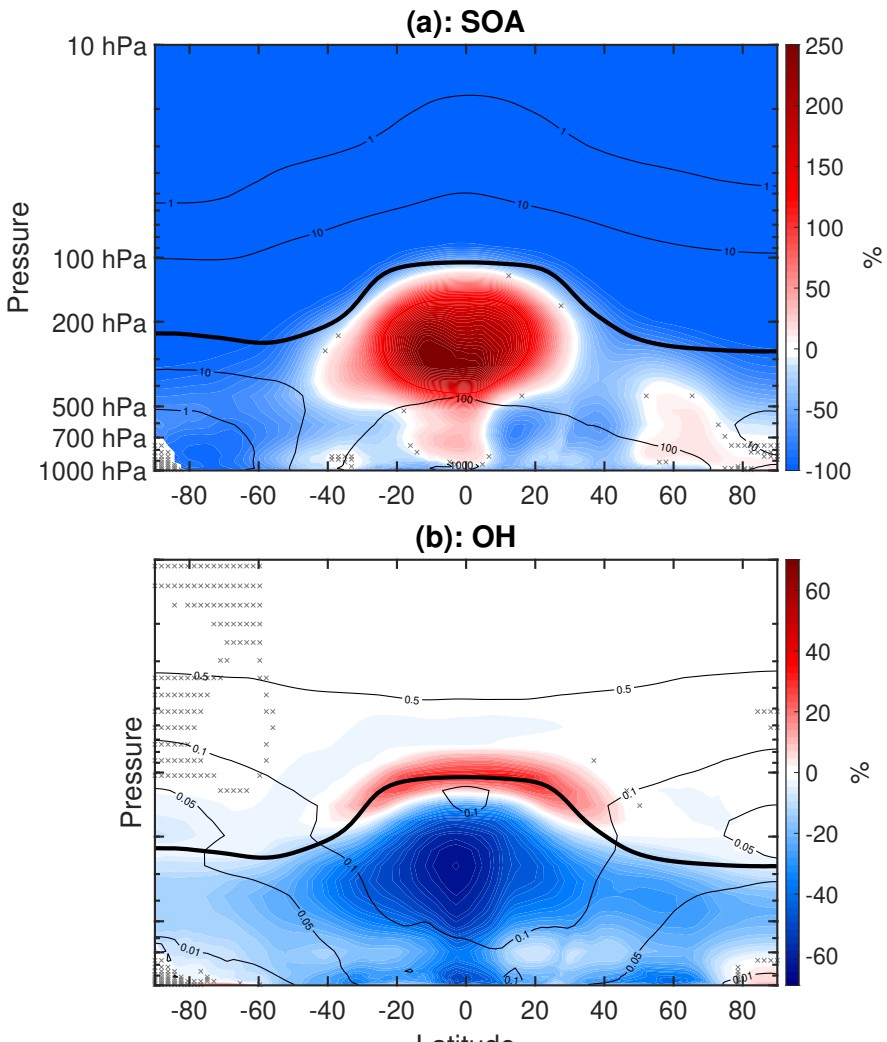

**Figure 10.** (a) Annual mean percent change (WACtl relative to MACtl) of secondary organic aerosol (SOA). The black contours indicate the MACtl climatology of SOA, with contours 1, 10, 100 and 1000 ng m$^{-3}$. (b) Annual mean percent change of OH. The black contours indicate the MACtl climatology of OH, contours 0.01 pptv, 0.05 pptv, 0.1 pptv, and 0.5 pptv. Gray crosses indicate responses that are statistically insignificant, with uncrossed responses being statistically significant at the 95% level. The thick black line marks the MACtl climatology of the tropopause.

the troposphere, there are SOA reductions of up to 80%, and nearly all of the stratospheric SOA is eliminated. We generally expect SOA increases as reactions are enabled between VOCs and SOA precursors. The reasons why SOAs decrease in some regions requires more detailed investigation of changes in SOA precursors and their possible consumption by other chemical reactions, and we leave such investigation for future work. Hodzic et al. (2016) also noted a reduction in stratospheric SOAs

due to fully interactive tropospheric chemistry. The SOA responses in our simulations do not offer additional direct insight into the shortwave heating responses, since as mentioned above, SOAs can have positive or negative effects on shortwave heating depending on the internal mixing. Nonetheless, Fig. 10a provides an indication of changes connected with numerous chemical pathways with potential climate effects that can be explored in the future.

The VBS mechanism for SOA formation would be expected to act as a sink for various radical species (OH, $NO_3$, NO, $NO_2$
and Cl). (To establish this effect with greater certainty, we would need to perform additional simulations that isolate the VBS mechanism, which is left for future work.) Reductions in these species can influence the HOx, NOx, and ClOx catalytic cycles, which in turn influence ozone formation (Finlayson-Pitts and Pitts Jr, 1999). In particular, the reduction of tropospheric OH concentration due to the inclusion of isoprene chemistry has been well documented in the literature (e.g., Von Kuhlmann et al., 2004; Squire et al., 2015; Bates and Jacob, 2019). To further illustrate just this one part of the complex web of interactions, Fig.
10b shows the change in OH concentration. This plot reveals an especially dramatic decrease in OH aligned with the strong increase in SOA in the tropical upper troposphere, which is expected since increased production of SOAs requires increased consumption of OH.

Increased emissions of VOCs (particularly isoprene) have been thought to increase the chemical loss of hydroxyl in the atmospheric boundary layer in low-NOx conditions (Von Kuhlmann et al., 2004). These initial hypotheses were contradicted
by observational studies in tropical forests which observed elevated OH concentrations, suggesting a previously-overlooked OH production term (e.g., Carslaw et al., 2001; Lelieveld et al., 2008; Martinez et al., 2010; Pugh et al., 2010; Whalley et al., 2011; Stone et al., 2011). In order to reconcile these differences, modellers looked to incorporate mechanistic OH-recycling in low-NOx conditions, which included isoprene epoxydiol formation, radical propagation in $HO_2$-acylperoxy reactions, and H-shift isomerizations of isoprene-hydroxy-peroxy radicals (Bates and Jacob, 2019). Archibald et al. (2010) found that the
H-shift isomerizations were an important mechanism for maintaining raised concentrations of OH in low-NOx conditions, with additional OH-recycling mechanisms being identified in subsequent studies (e.g., Bates and Jacob, 2019). However, it has been shown that the instruments used in some of the previous observational studies may have produced a positive OH bias due to internally generated OH and oxidation of biogenic VOCs (Mao et al., 2012). Due to its short lifetime and high reactivity, the details of OH chemistry remain highly uncertain, and thus OH is represented in numerous ways across 3-D CTMs. More
recently, Bates and Jacob (2019) found that the inclusion of isoprene-oxidation in GEOS-Chem decreased the global mean concentration of OH by 11%, with the largest decreases in the tropical continental boundary layer. The meridional structure of their OH changes qualitatively resembles that in our Fig. 10b, which suggests that inclusion of isoprene in WACtl leads to a reduction in OH compared to MACtl. Bates and Jacob (2019) also found that decreased OH increases the lifetime of $CH_4$ and appears to impact the $SO_2$-sulfate aerosol budget, as discussed further below.
BC has been classified as an AA, and it is treated as such in CESM2-WACCM6. BC can absorb incoming shortwave radiation and burn off clouds, a process known as the semi-direct effect. BC is thought to contribute to climate warming, though the exact magnitude of this effect is still uncertain (Menon et al., 2002; IPCC, 2021). Figure 11a shows the mean difference in BC, with blues indicating decreases and reds indicating increases. The MACtl climatology of BC (contours) peaks in the tropics and the NH midlatitudes, indicative of the rapid industrialization already occurring in Europe and North America during 1850. Fig. 11

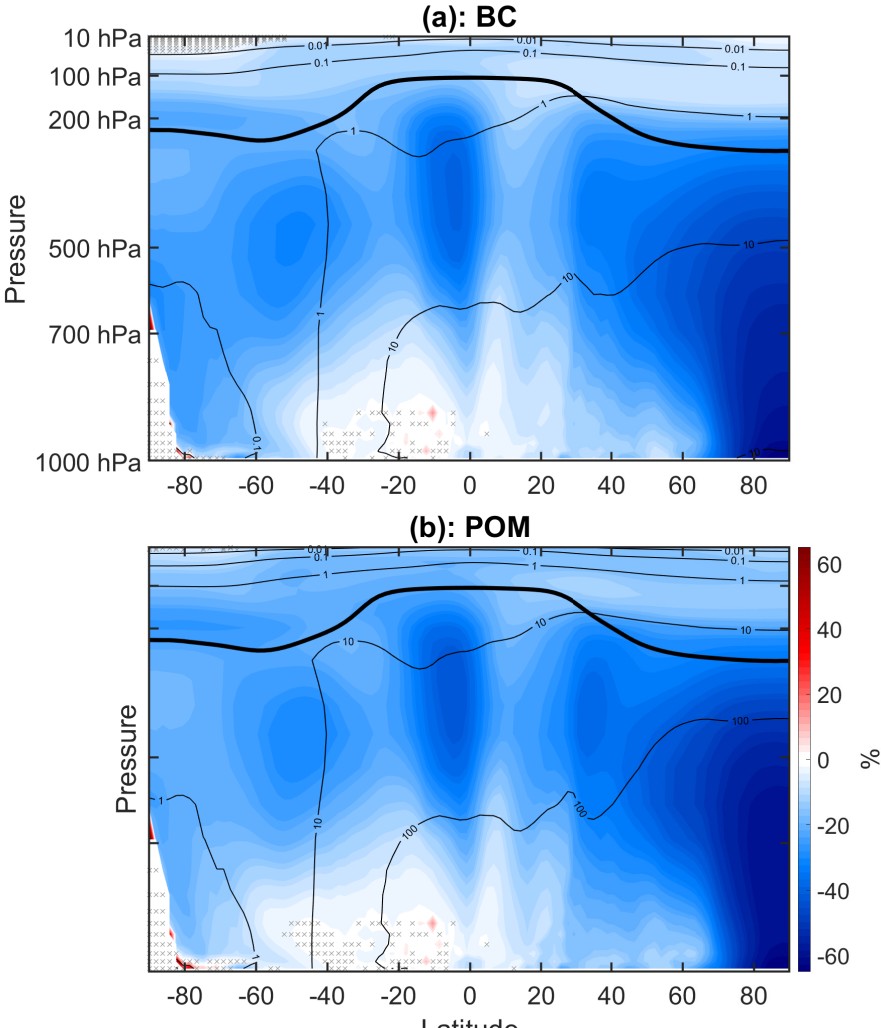

**Figure 11.** (a) Annual mean percent change (WACtl relative to MACtl) of black carbon aerosol (BC). The black contours indicate the MACtl climatology of BC, with contours 0.01, 0.1, 1, and 10 ng m$^{-3}$. (b) Annual mean percent change of primary organic matter aerosol (POM). The black contours indicate the MACtl climatology of POM, with contours 0.01, 0.1, 1, 10 and 100 ng m$^{-3}$. Gray crosses indicate responses that are statistically insignificant, with uncrossed responses being statistically significant at the 95% level. The thick black line marks the MACtl climatology of the tropopause.

shows decreased BC throughout most of the atmosphere, with the largest decreases (20-60%) occurring in the NH extratropical troposphere. We would expect such a decrease to contribute to the reductions in shortwave heating shown in Fig. 9, especially in the NH troposphere.

POM, in contrast to BC, is not strictly an AA. This ambiguity is due to the definition of POM. POM is considered aerosol that is directly emitted from diesel and gasoline exhaust as well as biomass burning (both anthropogenic and natural). POM can act as nuclei for SOA, or POM can become internally mixed with BC. POM can also be internally mixed with sulfate aerosol, which typically scatters more than it absorbs. So depending on the aging that occurs, POM can act more like an AA or more like a scattering aerosol (Song et al., 2007). In our simulations, however, the spatial distribution of POM in MACtl (Fig. 11b) closely corresponds with that of BC (Fig. 11a), and the WACtl – MACtl reductions in POM (30-65%) also closely correspond with those of BC. This correspondence suggests that, in our simulations, POM is internally mixed with BC and acts primarily as an AA. Thus, we can expect that the reduction in POM will also contribute to reductions in shortwave heating, especially in the NH troposphere.

Tilmes et al. (2019) also noted similar decreases in BC and POM when comparing fixed SST CESM2-WACCM6 simulations with and without explicit VOC chemistry. [The similarity of our results with those of Tilmes et al. (2019) is especially clear when comparing absolute changes in BC and POM, not shown.] This similarity suggests that these changes are ultimately not due to surface temperature changes. Rather, Tilmes et al. (2019) links these changes to overall increased SOA formation, which results in increased internal mixing of POM and BC with SOA, increased aging of POM and BC into the hydrophilic accumulation mode, and greater wet deposition of BC and POM in simulations with VOC chemistry compared to simulations with the SOAG scheme. Thus, even though BC and POM are not directly influenced by VOC chemistry, the effect of VOC chemistry on SOA production has indirect effects on BC and POM levels. Zheng et al. (2021) show, however, that MAM4 overestimates BC mixing with other aerosol types, so it is possible that the BC and POM responses in Fig. 11 are exaggerated, and additional work is needed to assess the indirect effects of VOC chemistry on BC and POM.

Increases in sulfate aerosol (primarily from volcanic eruptions, with smaller contributions from biomass burning and fossil fuel emissions) can lead to surface cooling, while decreases in sulfate aerosol can lead to surface warming (Martin et al., 2014; Schult et al., 1997). There are also biogenic contributions to sulfate aerosol, primarily dimethyl sulfide (DMS) from marine sources (Ghahremaninezhad et al., 2016). Figure 12 shows the mean difference in sulfate aerosol, with reds indicating increases and blues indicating decreases. Sulfate increases (up to 15%) in the SH extratropical upper and lower troposphere. Elsewhere in the troposphere, sulfate decreases with an especially pronounced decrease of approximately 35% in the tropical upper troposphere. This reduction somewhat aligns with the pronounced increase in SOA shown previously (Fig. 10a), suggesting that some of the sulfate aerosol reduction is due to internal mixing with SOA. This influence of internal mixing with SOA has also been noted by Tilmes et al. (2019), and this also influences the BC and POM changes as discussed above. The tropical sulfate changes appear to have an even closer alignment with OH changes (Fig. 10b). Note especially the OH increase at the tropical tropopause and the OH decrease in the tropical upper troposphere that align with qualitatively similar changes in sulfate. This correspondence suggests that OH changes are the dominant driver of tropical tropospheric sulfate changes, and we consider this possibility further below.

Sulfate aerosol is essentially unchanged in the tropical and NH lower stratosphere. Elsewhere in the stratosphere, there are sulfate increases ranging from about 3-8%. These increases approximately align with reductions in SOA (Fig. 10a), suggesting that reduced internal mixing with SOA is contributing here. However, other contributions, such as changes in the stratospheric

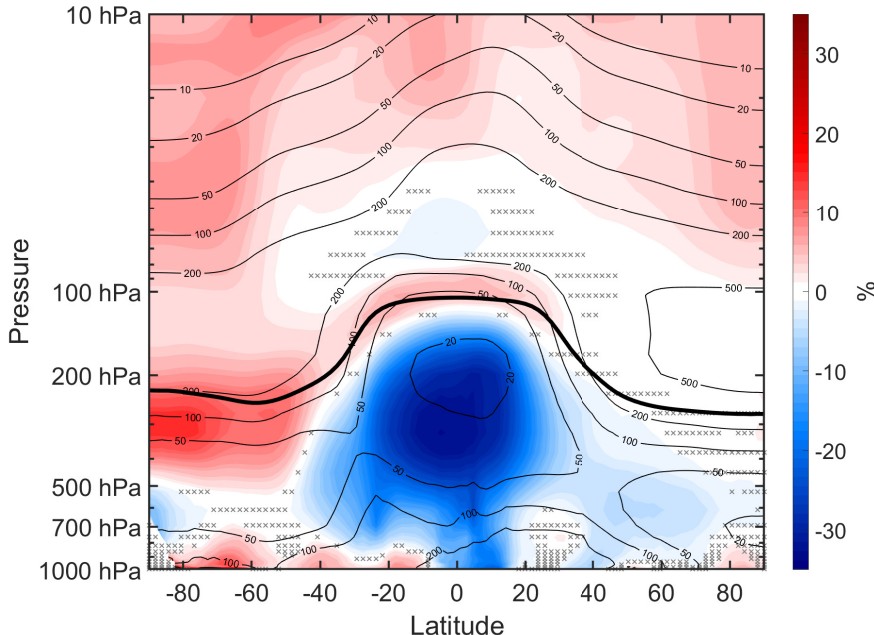

**Figure 12.** Annual mean percent change (WACtl relative to MACtl) of sulfate aerosol. The black contours indicate the MACtl climatology of sulfate aerosol, with contours of 10, 20, 50, 100, 200, and 500 $ng\,m^{-3}$. Gray crosses indicate responses that are statistically insignificant, with uncrossed responses being statistically significant at the 95% level. The thick black line marks the MACtl climatology of the tropopause.

circulation, need to be explored further, especially in the tropical and NH lower stratosphere, where sulfate is essentially unchanged despite the decreased SOA. We would expect the sulfate increases in the extratropical stratosphere to further contribute

to cooling in the extratropical troposphere. Furthermore, the sulfate reductions in the tropical troposphere would, on their own, lead to warming in the tropical troposphere. Thus, it is possible that, in the tropical troposphere, warming due to decreased sulfate aerosols is opposing cooling due to changes in other constituents (e.g. BC, POM), resulting in nearly unchanged temperatures in the tropical troposphere (Fig. 6). As mentioned above, verifying such a possibility will require additional controlled experiments which we plan to perform in the future.

Fig. 13a shows the zonal mean percent change in the vertical column density (VCD) of sulfate aerosol burden. These changes reiterate key features apparent in Fig. 12, showing that vertically-integrated sulfate decreases in the tropics and increases in the extratropics. We have performed additional analysis to assess the roles of OH changes and wet deposition. The primary chemical sink of $SO_2$ is due to a reaction with OH, a process that eventually produces sulfate aerosol (Lachatre et al., 2022). Sulfate can be removed from the atmosphere via wet and dry deposition (Liu et al., 2001; Zhang et al., 2001), and its abundance

is also influenced by internal mixing with SOA as discussed above. Therefore, if there were no changes in $SO_2$, SOAs or wet deposition, a decrease in OH would result in a decrease in sulfate.

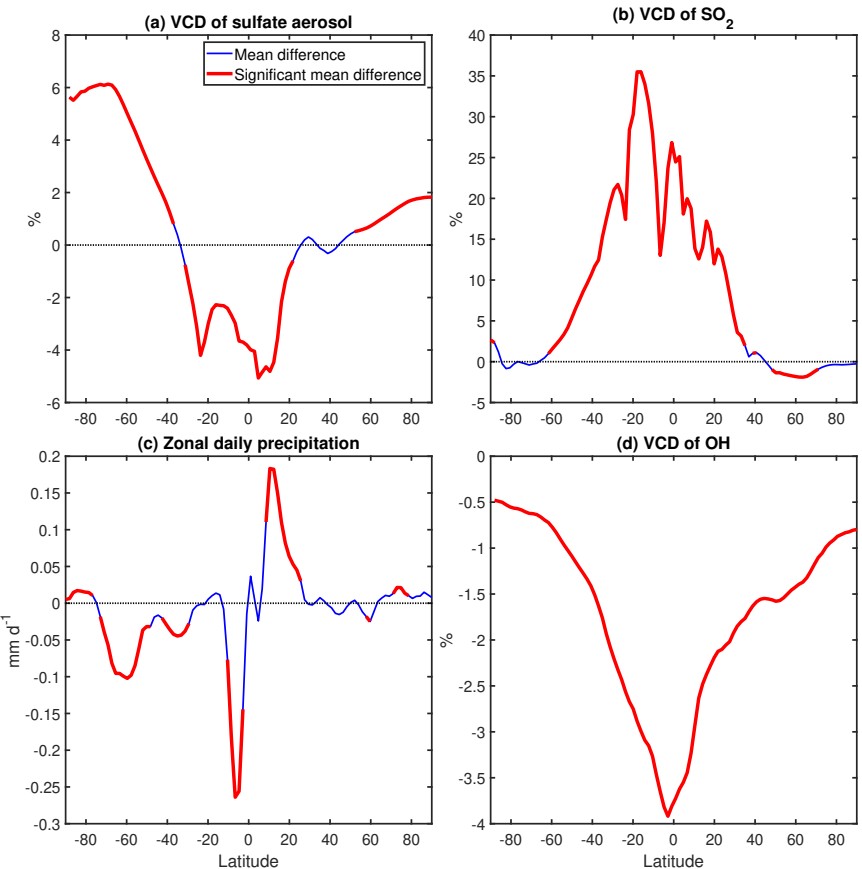

**Figure 13.** (a) Annual mean percent change (WACtl relative to MACtl) in vertical column density (VCD) of sulfate aerosol. (b) Annual mean percent change in VCD of $SO_2$. (c) Annual mean change of zonal mean precipitation ($\mathrm{mm\,d^{-1}}$). (d) Annual mean percent change in VCD of OH. The differences that are statistically significant at the 95% level are colored red.

Fig. 13b shows that the VCD of $SO_2$ increases from approximately 60°S to 40°N, whereas Fig. 13d shows that the VCD of OH decreases over all latitudes, with the strongest OH decreases in the tropics. This suggests (as was apparent from our earlier comparison of Figs. 10b and 12) that the tropical sulfate aerosol decreases are primarily due to decreases in OH. These decreases appear to be partially diminished by a precipitation decrease south of the equator (implying less wet deposition) and enhanced by a precipitation increase north of the equator (implying increased wet deposition; Figs. 13a,c). This result indicates that wet deposition changes are also contributing to sulfate aerosol changes, but these changes appear to be secondary to the chemical influence of OH changes. As discussed above, tropical SOA increases (Fig. 10a) might also be contributing to the tropical sulfate decreases. Furthermore, the most plausible explanation for increased sulfate aerosol outside of the tropics is the decreased extratropical SOA, as $SO_2$ and OH changes are weak in the extratropics. The sulfate aerosol increase in the SH extratropics appears to be further enhanced by a decrease in precipitation over approximately 30-70°S.

## 6 The role of zonal mean cloud changes

In addition to the changes in shortwave forcers mentioned in the previous section, changes in clouds are also expected to contribute to changes in shortwave heating. In Section 3, we showed evidence that cloud feedbacks were influencing surface temperature changes over widespread regions. Here, we explore these changes further from the zonal mean perspective to assess possible contributions to zonal mean temperature changes.

Figure 14 shows the zonal mean differences in net CRE (panel a) along with the zonal mean changes in cloud fraction (panel b). (In panel b, reds indicate increases in cloud fraction and blues indicate decreases.) There is an overall downward shift in clouds over most latitudes. Such a downward shift is expected, as cooler temperatures would result in condensation occurring at lower altitudes. Such a downward shift is qualitatively opposite to the upward shift seen in global warming simulations (e.g., Tandon and Cane, 2017). The extent to which changes in cloud condensation nuclei (CCN) might also be contributing to these cloud changes needs to be further investigated. The changes around 60°S are more vertically aligned, suggesting that dynamical changes, such as the equatorward shift of the midlatitude jet, are playing a role. Indeed, earlier studies have shown the strong connection between clouds and the response of the SH midlatitude jet to climate forcing (e.g., Grise and Polvani, 2014; Ceppi and Hartmann, 2016). In the SH midlatitudes and NH high latitudes, the downward shifts of the clouds result in CRE decreases as expected. As the clouds shift equatorward with the SH midlatitude jet, there is a sharp increase in CRE around 60°S, on the poleward flank of the midlatitude jet. Around this latitude, there is also a reduction in sea salt aerosol associated with the reduced near-surface wind speeds that appears to drive a reduction in cloud liquid (not shown) and amplify the cloud fraction decrease/CRE increase. Near the equator, the downward shift in upper-level clouds appears to be counteracted by a reduction in mid-troposphere clouds, resulting in a CRE increase here.

Comparing the cloud fraction changes with the shortwave heating changes (Fig. 9), we see a number of locations where the changes correspond with each other, such as in the tropical middle-to-upper troposphere, and around 50°S (in the vicinity of the SH midlatitude jet). This correspondence suggests that cloud changes are driving some of the shortwave heating changes. The apparently cloud-driven shortwave heating increase spread over the tropical upper troposphere is of particular interest because idealized studies have shown that such heating on its produces HC weakening and expansion (e.g., Tandon et al., 2013). Thus, the complexity in the circulation response due to VOC chemistry may be due to the competing effects of extratropical shortwave forcers and tropical cloud feedbacks. Widespread extratropical cooling (possibly due to changes in sulfate aerosols, BC and POM) would be expected to shift the midlatitude jets equatorward, whereas tropical increases in shortwave heating (likely due to tropical cloud changes) would result in HC weakening and expansion.

## 7 Conclusions

This study examines the climate effects of tropospheric VOC chemistry by taking the difference between two simulations of CESM2-WACCM6: 1) WACtl, a case with explicit VOC chemistry, and 2) MACtl, a case with a simplified SOAG scheme and no explicit VOC chemistry. Changes (WACtl – MACtl) in surface temperature range between -4 K and 4 K. This result motivates further research to investigate the possible influence of VOC chemistry on anthropogenic climate change. While we

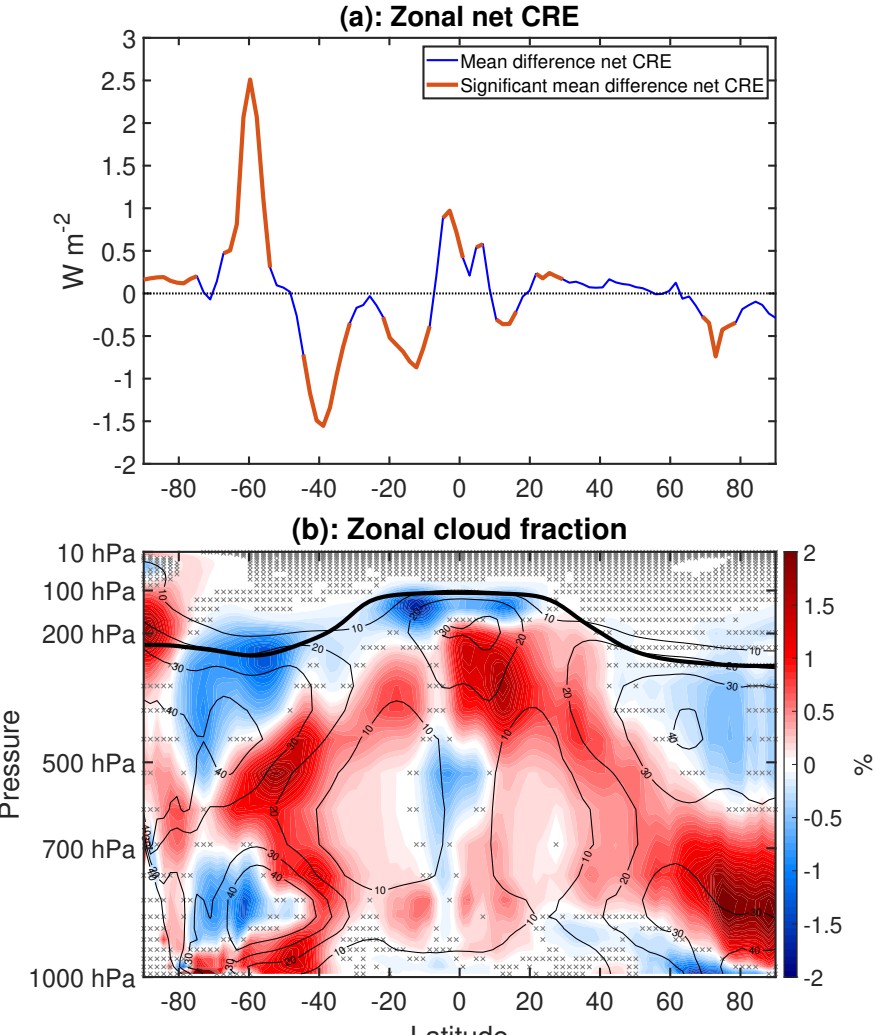

**Figure 14.** (a) Annual mean difference (WACtl – MACtl) in net cloud radiative effect (W m$^{-2}$). The differences that are statistically significant at the 95% level are colored red. (b) Annual mean difference (WACtl – MACtl) in zonal mean cloud fraction (%). The black contours indicate the MACtl climatology of cloud fraction, with contours 10%, 20%, 30%, and 40%. Gray crosses indicate responses that are statistically insignificant, with uncrossed responses being statistically significant at the 95% level. The thick black line marks the MACtl climatology of the tropopause.

explored the mechanisms of the large-scale temperature changes, our simulations also show significant localized temperature changes over land. The reasons for these localized changes are unclear, and this is an important topic for follow-up study. Such investigation would require analysis of regional changes in shortwave, longwave and dynamical heating, going beyond our zonally averaged analysis of these quantities.

In the zonal average, there is widespread tropospheric cooling in the extratropics. This cooling results in an equatorward shift of the SH midlatitude jet that is in turn associated with shifts of precipitation patterns and cloud cover. This cloud shift results in a positive cloud feedback in the SH high latitudes that offsets cooling there. Elsewhere in the extratropics, the overall cooler temperatures (possibly in combination with CCN changes) produce a downward shift of clouds that produces additional cooling in the midlatitudes. There is also weakening and expansion of the HCs that appears to be due to a cloud-driven shortwave heating increase in the tropical upper troposphere. This HC weakening drives additional precipitation changes in the tropics and subtropics.

We presented evidence that changes in longwave forcers were likely not the root cause of these climate responses. Instead, there is evidence that shortwave forcers, such as decreased BC, decreased POM and increased stratospheric sulfate aerosols are contributing to the extratropical cooling. In the tropics, there is evidence that reductions in sulfate aerosols offset the cooling due to other shortwave forcers, resulting in nearly unchanged tropical temperatures. Confirming the roles of these shortwave forcers will require performing additional controlled experiments, which we plan to do in the future.

As found in Tilmes et al. (2019), slower SOA formation allows for increases in SOA that are especially dramatic in the tropical upper troposphere. This overall SOA increase results in increased internal mixing of sulfate aerosol, BC, and POM with SOA, which strongly influences the changes in these constituents. Furthermore, SOA decreases throughout the stratosphere, which appears to drive the sulfate increase over most of the stratosphere. Thus, the effects of VOC chemistry extend to aerosols that do not directly participate in VOC chemistry.

Our simulations also produce significant changes in tropospheric and stratospheric ozone (Fig. A2). We presented evidence suggesting that these ozone changes likely do not play a strong role in the large-scale climate response, as the strong warming in the SPS is mostly driven by dynamical heating. Nonetheless, we plan to perform additional controlled model experiments with prescribed ozone to confirm that ozone changes are playing a minor role. Furthermore, the mechanisms driving these ozone changes are interesting in their own right and deserving of further study.

There are important limits to the representation of chemistry in CESM2-WACCM6, such as the exclusion of HONO. Furthermore, for simulations that include recent anthropogenic emissions (which our simulations exclude), there are additional species absent from CESM2-WACCM6, such as methyl nitrate and peroxypropionyl nitrate (PPN) that might be important to consider. Thus, additional work is needed to improve the representation of chemistry in CESM2 and develop confidence that the responses we obtain capture the effects of VOC chemistry in the real world. As such improvements are made, maintaining a hierarchy of models with both coupled and uncoupled chemistry will enable continued efforts to disentangle the numerous possible climate effects of tropospheric chemistry. In addition to CESM2 improvements, we look forward to additional analysis of existing simulations, additional simulations with other models, and additional observational work that can help to assess the generality of our findings.

*Code and data availability.* The code for CESM2 is avaiable through github via the instructions given on NCAR's CESM2 web portal: https://www.cesm.ucar.edu/models/cesm2. Output data analyzed in this study are available from the authors upon request.

## Appendix A

Here we show some additional figures that support some points in the main text, but are beyond the main focus of our study.

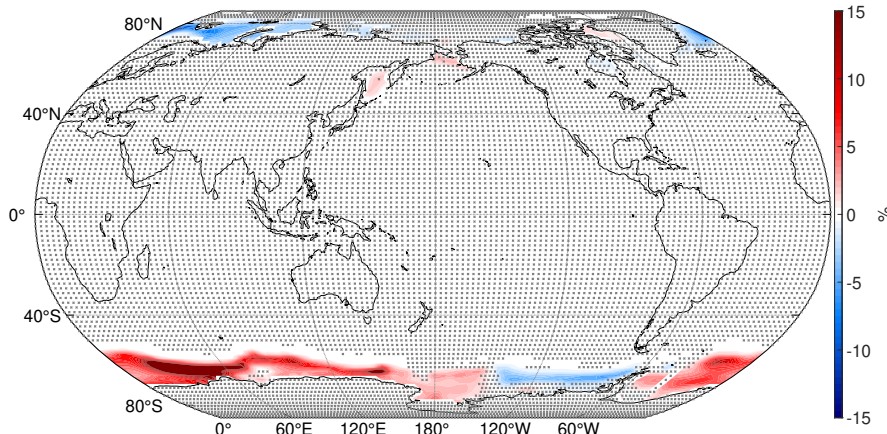

**Figure A1.** Annual mean difference (WACtl – MACtl) in sea ice concentration (%). Gray crosses indicate responses that are statistically insignificant, with uncrossed responses being statistically significant at the 95% level.

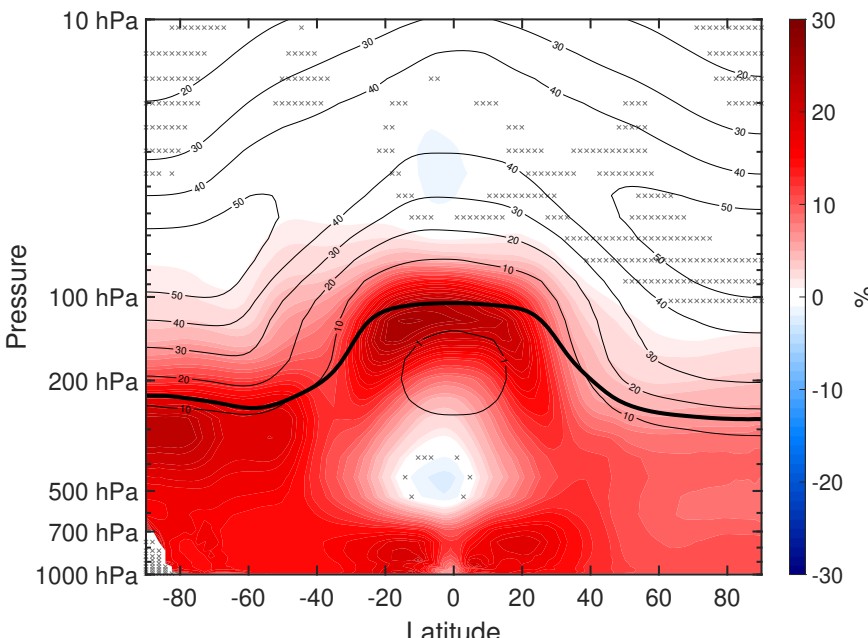

**Figure A2.** Annual mean percent change (WACtl relative to MACtl) in ozone. The black contours indicate the MACtl climatology of ozone, contours $1 \times 10^{12}$ molecules cm$^{-3}$. Gray crosses indicate responses that are statistically insignificant, with uncrossed responses being statistically significant at the 95% level. The thick black line marks the MACtl climatology of the tropopause.

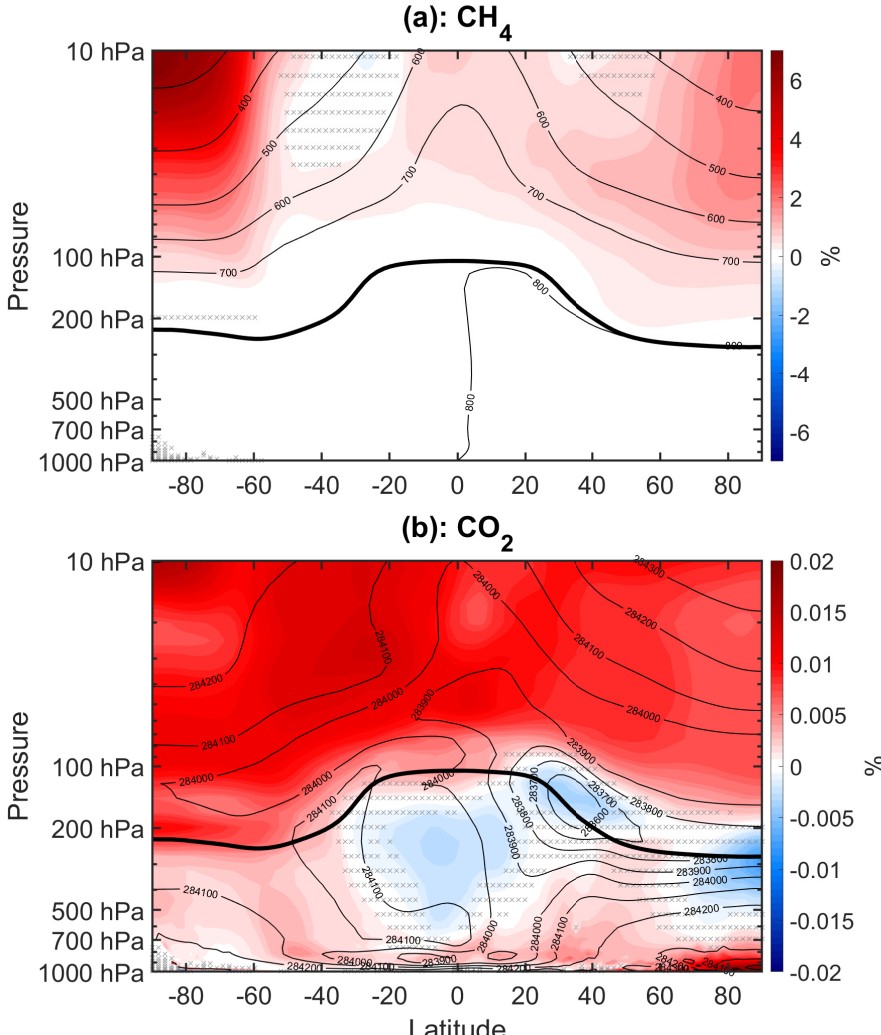

**Figure A3.** (a) Annual mean percent change (WACtl relative to MACtl) in CH$_4$. The black contours indicate the MACtl climatology of methane, with a contour interval of 100 ppbv. (b) Annual mean percent change in CO$_2$. The black contours indicate the MACtl climatology of CO$_2$, with a contour interval of 100 ppbv. Gray crosses indicate responses that are statistically insignificant, with uncrossed responses being statistically significant at the 95% level. The thick black line marks the MACtl climatology of the tropopause.

*Author contributions.* N.A.S. performed the model experiments and analysis and led the writing of the manuscript. N.F.T. provided guidance on the model experiments and analysis, and he edited the manuscript.

*Competing interests.* The authors have no competing interests.

*Acknowledgements.* Special thanks to Rob McLaren, Yongsheng Chen and Cora Young for helpful discussions as well as suggestions on an earlier version of this manuscript. We also thank Jamie Ward, Mike Mills, Simone Tilmes and Andrew Gettelman for helpful discussions and correspondence. The editor (Kostas Tsigaridis), Alexander Archibald and an anonymous reviewer provided very valuable feedback on the submitted manuscript. Computing resources were provided by the Digital Research Alliance of Canada. The CESM project is primarily supported by the U.S. National Science Foundation.

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
