# Peer review of "How Does Tropospheric VOC Chemistry Affect Climate? An Investigation of Preindustrial Control Simulations Using the Community Earth System Model Version 2"

_Atmospheric Chemistry and Physics, 2023_

## Author Comment (AC2)

We sincerely thank Alexander for his valuable feedback that will help to improve our manuscript. Below we provide point-by-point responses (in red) to Alexander's comments (reproduced in black italics), with modifications to the manuscript indicated in **bold**.

*Stanton and Tandon present numerical model calculations, which aim to attribute the role of Volatile Organic Compounds (VOC) on climate. This is a worthwhile question and progress on the topic is needed inlight of efforts like the WCRPs Lighthouse Activity on "Explaining and Predicting Earth System Change". However, this study falls significantly far from being able to answer the question. I think the authors should be comended for their attempt with the modelling but significant flaws in the model design mean that they are unable to answer the problem they have set out to. I would suggest a significant revision, including a change of the title of the manuscript if the present results are to be used as the underpinning data for a revision.*

*Good points:*

*Overall the manuscript is well written. The climate analyses are excellent and the team are clearly experts in this area.*

Thank you for this positive feedback.

*Bad points:*

*The experiment design is very much flawed in being able to answer the question of VOCs impacts on climate. We have known for a long time that the effect of VOCs is non-linear on key climate forcers (e.g., O3, CH4, aerosols). By removing the VOCs one is significanlty perturbing the system into a state which does not allow the role of VOCs to be teased out. The study answers the question: "What are the impacts of removing VOC chemistry on pre-industrial climate simulated by CESM2". This is a useful question but far from the grander question raised by the title. I think this point is a critical one that can not be addressed without addressing the core of the study/revising the aims and scope.*

Thank you for this feedback. We agree that the influence of VOC chemistry would depend on levels of other atmospheric constituents like ozone, methane and aerosols. For this reason, we do not think it would be feasible in one paper to answer the general question of how VOC chemistry influences climate across all possible climate states. For the question to be well-posed, the base climate state needs to be specified. We make clear in the abstract and text that we focus only on preindustrial simulations, which is such a standard assumption for a base climate state that we didn't think it was necessary to specify that in the title as well. But prompted by your feedback, we will modify our title to be "How Does Tropospheric VOC Chemistry Affect Climate? An Investigation **of Pre-Industrial Control Simulations** Using the Community Earth System Model Version 2." This hopefully makes it absolutely clear that we are looking at just preindustrial climates.

As for your comment that "By removing the VOCs one is significanlty perturbing the system into a state which does not allow the role of VOCs to be teased out." We agree that removing VOCs entirely would be a potentially extreme perturbation to the chemistry-climate system, with a whole chain of possible effects that would likely be hard to disentangle. But it would be

misleading to say that VOCs are absent in our MACtl simulation. Rather, in MACtl, the chemical reactions involving VOCs are absent, and they are replaced by the SOAG scheme (Liu et al., 2012; Emmons et al., 2010). So the difference between WACtl and MACtl is not the presence of VOCs but rather the presence of VOC *chemistry*. We will add text to Section 2 of the manuscript to make this clear.

*As someone who has looked at aspects of the title problem I am very surprised by the large changes in surface temperature that the authors show in Figure 1. The small difference in Global Mean Surface Temperature (GMST) suggests that the surface response is really not that signficiant. How does that compare to the spread of PI GMST simulated by CMIP6 models or even ensemble members of CESM2? Are these changes really significantly different from the uncertainty in the literature? The model we use in the group, UKESM-1-LL, shows significant variance across ensemble members (even under PI conditions). As it currently stands, I don't think the results suggest that there is a significant imapct of VOC chemistry on climate; BUT as I said I am not sure that the experiment design allows one to answer this question.*

Thank you for this feedback. Hopefully, our response to your previous comment clarifies our experimental setup's relevance to the scientific question. Given that SOA effects can be highly regional, we should not necessarily expect a large GMST response to VOC chemistry. While our statistical significance tests provide some confidence that our surface temperature responses are significant, we can show some additional results to increase confidence in the robustness of our results. First, we have extended our simulations by 70 years so that the averaging period is now 211 years. Our revised manuscript will show results from these extended simulations, and the updated Fig. 1 is reproduced below as Fig. R2-1. All of the key features in these results are the same as in the submitted manuscript, indicating that internal variability was not strongly influencing our results.

[Figure]

Figure R2-1: Annual mean difference (WACtl – MACtl) in surface temperature (K) from 211 years of model output. Gray crosses indicate regions that are not statistically significant, with uncrossed regions being statistically significant at the 95% level.

We do not think it would be particularly helpful to look at the spread among different CMIP6 models, as that spread can be due to intermodel differences in climate states, not just internal

climate variability. Furthermore, in the CESM2 large ensemble, the atmospheric component was the Community Atmospheric Model (CAM), not WACCM, so it would not cleanly compare with our simulations. The CMIP6 archive does include a preindustrial control simulation using the same configuration as our WACtl simulation, but it is only 250 years long, which is not long enough to be helpful for this discussion. However, the CMIP6 archive also includes a 500-year preindustrial control simulation using the same configuration as our MACtl simulation. This is long enough to give an indication of internal variability when averaging over time periods comparable to the lengths of our simulations.

Below is the difference in surface temperature when averaging over non-overlapping 211-year time slices in this CMIP6 control run of CESM2-WACCM6. Each of the panels shows a different choice of averaging period, and they are plotted on the same shading scale as used in Fig. 1 of our manuscript and Fig. R2-1 above. None of these calculations shows temperature differences comparable to the significant temperature differences shown in Fig. R2-1. This analysis further establishes that the responses shown in our simulations are not due to internal climate variability, and the responses represent the preindustrial climate effects of VOC chemistry. We will add this analysis to the manuscript.

[Figure]

Figure R2-2: Difference of time-averaged surface temperature between non-overlapping 211-year time slices of the CMIP6 preindustrial control simulation of CESM2-WACCM6-FV2 (variant label r1i1p1f1). This simulation uses the same configuration as our MACtl simulation. The precise simulation years that were averaged are indicated in the title over each panel.

*What was also lacking for me was more of a focus on the causal links between tropospheric composition changes and their impacts on radiative forcers. There is clearly a very large response in OH. Why? I would guess the removal of isoprene, which has been addressed before, many times (see e.g., Bates and Jacob, 2019; Squire et al., 2015; von Kuhlmann et al., 2004). I was surprised not to see mention of isoprene at all in any analyses. Moving from OH one can*

*then identify the impacts of changes in oxidising capacity on aerosols and aerosol precursors. What happens to SO2? Why does Sulfate change the way it does? Some of this can only be understood by constructing budgets of the variables (production and loss) and analysing them. Like I said the climate analysis is good but the attribution to composition changes leaves a lot to be desired and can be thoroughly improved to provide insight into causality (but not attribution of it!).*

Thank you for this feedback. Just to clarify, isoprene chemistry is included in WACtl but not in MACtl. Yes, the addition of isoprene chemistry in WACtl likely explains the large decrease in OH relative to MACtl. We will add details of the impacts of isoprene chemistry in the results section under figure 10 along with references to the earlier studies you mention (Bates and Jacob, 2019; Squire et al., 2015; von Kuhlman et al., 2004). For example, L485 "**The reduction of tropospheric concentration of OH due to the inclusion of isoprene chemistry has been well documented in the literature (e.g., Bates and Jacob, 2019; Squire et al., 2015; von Kuhlman et al., 2004). To illustrate this link in the chain, …**". The influence of OH versus precipitation on $SO_2$ and sulfate changes will be addressed in a comment below.

*In additio to these minor comments I have more minor comments:*

*L26: The abstract is missing a conclusive statement at the end.*

The following statement will be added to the end of the abstract: "**Some of these responses are quantitatively large enough in some regions to motivate investigation of possible influences of VOC chemistry on anthropogenic climate change.**"

*L38: I don't think oxidization is a word. Change to oxidation.*

Thank you for catching this, we will correct this to "**oxidation.**"

*L41: Not clear how RO2 makes NO3. Add a reaction or reference. Need to define HO2 and RO2.*

Sorry for the confusion here. We should have said $RONO_2$ instead of $NO_3$ here. The manuscript will be modified here as follows: "These peroxy radicals **($RO_2$)** can participate in further oxidation reactions, producing a range of products, such as **organic nitrates ($RONO_2$) and hydroperoxyl radicals (HO2)**."

*L45: Not only "typically". Actually. That's the nature of a CTM. See Young et al (2018) for an overview of the types of models and feedbacks/couplings and adopt that nomenclature.*

Thank you, we will change "is typically" to "**by design is**" and we will insert the CTM acronym here in order to adopt standard nomenclature.

*L56: They didn't have to prescribe SSTs. They chose to. there are still elements of climate response with fixed SSTs and the use of fixed SSTs is the defacto method for calculating key climate metrics like ERF.*

Thank you, we agree. The text here will be changed to the following: "Climate simulations with more comprehensive chemistry **typically prescribed sea surface temperatures (SSTs) (e.g.,**

**Tilmes et al., 2019), which facilitates calculating some climate metrics like effective radiative forcing (ERF) (e.g., Forster et al., 2016), but this approach** limits the ability…".

*L61: Define NOx.*

On L61 the following will be added after $NO_x$: "$(NO_x = NO + NO_2)$".

*L73: ...limits the model's ability to produce a "full" climate response. Add "full".*

Thanks, we will add "**full**" here.

*L84-85: Absolute temperature variations across climate models is way larger than this. Anyway, what is key to climate change is the difference between simulations under different forcings within a model. This statement can be misconstrued and so needs to be toned down, alot. I still don't see the surface temperature response as being at all significant so would like to be convinced more on this point.*

Our response to your earlier comment hopefully provides reassurance about the robustness of our results within the framework of CESM2-WACCM. (And we agree, the spread across CMIP6 models isn't the key consideration here.) We see that our statement here could be misconstrued to challenge the role of greenhouse gases in future climate change. So, we will modify the text here as follows: "When the effects of tropospheric VOC chemistry are isolated, our simulations show significant impacts on climate, comparable in some regions to the temperature changes **expected from greenhouse gas increases** over a century in future climate change scenarios."

*L87: Typically RONO2 are formed from RO2+NO too. Should be made clear if BVOC+NO3 is the only route to RONO2 or if there are other routes, too. And what Organic Nitartes are comprised of in the model. PAN?*

$BVOC + NO_3$ is not the only route for $RONO_2$ in the WACtl case. $RO_2 + NO$ is also included. PAN is included in WACtl, but not in MACtl. In WACtl, organic nitrates consists of PAN and a number of other species (Emmons et al., 2020). We will update the manuscript here to clarify this.

Based on further investigation of how NOy is treated in the model, we will be updating much of this section. For example, we recently learned that, while gas phase NOy is included in the model, the condensed phase (which impacts radiation) is lumped with sulfate aerosols. This matter will also be addressed in a comment below.

*Table 1: It's unclear how key reactions, like the CH4 oxidation, varies between experiments. Neither is it clear what the emissions of NOx and VOCs are.*

The $CH_4$ oxidation reactions are the same in WACtl and MACtl, with the only difference being a $CH_4$ production term in the WACtl case. This production term enters into the reaction of propene with ozone ($O_3 + C_3H_6 \rightarrow C_2H_4O_2 + HCHO$, $C_2H_4O_2 \rightarrow CH_4 + CO_2$).

The emissions of VOCs are addressed further down in the manuscript (see L189 to L193). NOx emissions are addressed on L184 and L193-194. This will be clarified in the revised manuscript.

*L143: So there is no methane? No CO? Why? NB I am not saying CO is a VOC.*

$CH_4$ and CO are solution species in both MACtl and WACtl. The manuscript will be updated here to make this clear: "any explicit **non-methane VOC (NMVOC)** chemistry in the troposphere,…"

*L175: I agree HONO is important but many things are missing that are important for ozone (see for example my review paper on tropospheric ozone, Archibald et al. 2020). The focus on HONO seems quite parochial.*

Thank you for this comment. A large majority of the species present in Table 1 of Archibald et al. (2020) are present in WACtl except HONO, $CH_3CH(OO)CH_3$, $CH_3CH(OOH)CH_3$, $C_2H_5CO$, $C_2H_5C(O)OO$, PPAN, $MeONO_2$ (assuming this is $CH_3ONO_2$), MSA, and DMSO. To our knowledge, besides HONO, none of these additional species are expected to be important for ozone. Note that in Table 1 of Archibald et al. (2020) there may be a typo. The variable named MGLY has the formula ($CH_3COCHHO$), but it appears this is meant to be methyl glyoxal, which has the formula ($CH_3COCHO$).

*L198: I think this is a major weakness of the study. I would welcome comments on the limitations that this imposes and whether 39 years is really long enough for a fully coupled model to "spin-up".*

Neither of our model simulations was "spun-up" from a rest state. Rather the WACtl case was initialized from a climatological ocean and sea ice state, and the MACtl run was initialized from a reference case of the same model configuration that was already spun up. As we showed in our response to the Anonymous Reviewer (see Fig. R1-1), the difference between the initial SST states appears to be a shift in the phase of internal variability (specifically El Nino-Southern Oscillation, ENSO), rather than a fundamental difference in the long-term climate state. So 39 years is a reasonable time period to allow for WACtl to adjust from its initial state, as that adjustment would mostly involve an adjustment to VOC chemistry. We will update the manuscript to make these points.

*Figure 1: This is rather puzzling a result. For a start the largest changes are over the oceans. This is where some fixed SST runs would help to remove any noise caused by changes in ocean circulation which will have a long time signal (and require more than 140 years of run). The main changes in temperature seem associated with sea ice zones. Is that correct? What could cause such large and significant ~ grid-box level changes in surface temperature over land? I can't think of a mechanism associated with chemistry.*

We thank you for this feedback. While running fixed SST simulations would reduce noise, it would greatly limit the full climate response, which is what we are interested in. Hopefully our response to your earlier comment addresses any concern that our responses are strongly influenced by internal variability. While the most significant surface temperature changes are in the extratropics, these changes are not confined to sea ice zones.

We are not sure of the mechanisms driving the localized surface temperature changes over land. Most of our investigation of mechanisms was focused on the zonal mean changes, and we plan to investigate more localized changes in future studies. We will update the manuscript to this effect.

*Figure 1-Figure 6 and results and discussion. How large are these changes compared to for example the spread of CMIP6 models and or the spread of LENS/2 simulations with CESM/2.*

Please refer to our earlier response and Figure R2-1 regarding the comparison of the surface temperature response to internal variability. As we stated before, comparisons to the spread of CMIP6 models and LENS simulations would not be especially conclusive here, but we provided comparison to a longer CMIP6 control simulation using CESM2-WACCM6.

*L445: Are RONO2 coupled to the radiation scheme?*

Thank you for this question. As we stated above, we investigated further and found out (through correspondence with the modelling centre) that gas phase $RONO_2$ is not coupled to the radiation scheme, and condensed phase $RONO_2$ is lumped with sulfate aerosol (which does impact radiation). So the radiative effects of condensed organic nitrates are not treated separately from those of sulfate aerosols. This updated understanding requires updates to our results sections, as we can now be more confident that the widespread cooling in the extratropical troposphere is likely due to increased sulfate aerosols in the stratosphere combined with reduced sulfate aerosols in the troposphere.

*Figure 9-12: % changes would be much more helpful. Please also express species in units that are more widley used in the literature. pptw for example is not used in atmospheric chemistry circles. Instead use mass per unit volume (g/m3 for example).*

Thank you for these suggestions. We did experiment with showing % changes, and we found that % changes can potentially be misleading in the results. This is because, in regions where the climatological value of a species is very low, then a very small absolute change can produce a huge percent change. We will update the manuscript in a few places where percent changes may be suitable (i.e., Figure 10, and 11). Any figures in pptw units will be updated to be in units of g m$^{-3}$ or mg m$^{-3}$. For gas-phase species, the usage of ppm still seems appropriate.

*L465: What matters more is how SOA are dealt with in the model, not what the literature says. Can you please expand on the coupling of SOA to radiation (through direct and indirect effects).*

We appreciate this feedback. SOA is indeed coupled to the radiation scheme (RRTMG) in CESM2-WACCM6 in both the WACtl and MACtl cases. This coupling is through both the direct effect (influencing aerosol optical depth) and the indirect effect (SOA acting as cloud condensation nuclei). Clouds are also coupled to RRTMG. This will be clarified in the methods and results sections.

*L483: This causality is not possible to determine. One would need to isolate ONLY the VBS to assert this.*

Thank you for pointing this out. The language of causality will be removed here. The manuscript will be changed as follows: "**Although causality cannot truly be determined,** the VBS

mechanism for SOA formation **would be expected to act as a sink for various radical species (OH, NO$_3$, NO, NO$_2$ and Cl). To verify this point, a sensitivity analysis of only the VBS mechanism is required, which is left for future work."**

*L523: This can be because: 1) you have less OH so less SO2 forms sulfate 2) you have more clouds and rain so more SO4 is wet deposited. Which is it? I think you need to examin the SO4 budget.*

We greatly appreciate these points as they have helped to strengthen our analysis. It appears that OH changes have a stronger influence compared to wet deposition changes. To provide evidence for this, a new figure will be included in the results section, reproduced below.

[Figure]

Figure R2-3: (a) Annual mean difference (WACtl – MACtl) in vertical column density (VCD) of sulfate aerosol (kg m$^{-2}$), (b) VCD of SO$_2$ (kg m$^{-2}$), (c) zonal mean precipitation (mm d$^{-1}$), and (d) VCD of OH (kg m$^{-2}$). The differences that are statistically significant at the 95% level are colored red.

Panel (a) indicates a decrease in sulfate aerosol over the tropics, where OH decreases, but there is no widespread increase in precipitation in the tropics. This suggests that the tropical decreases in

sulfate are driven primarily by changes in OH rather than changes in wet deposition of $SO_4$. It is notable, however, that over approximately 2-25ºN, there is a precipitation increase that aligns with a local minimum in the sulfate aerosol change. This suggests that increased wet deposition of $SO_4$ is also contributing to the sulfate aerosol decrease in particular regions.

References:

Bates, K.H, and Jacob, D.J.: A new model mechanism for atmospheric oxidation of isoprene: global effects on oxidants, nitrogen oxides, organic products, and secondary organic aerosol, Atmospheric Chemistry and Physics, 19, 9613-9640, https://doi.org/10.5194/acp-19-9613-2019, 2019.

Emmons, L.K., Walters, S., Hess, P.G., Lamarque J.-F., Pfister, G.G., Filmore, D., Granier, C., Guenther, A., Kinnison, D., Laepple, T, et al.: Description and evaluation of the Model for Ozone and Related chemical Tracers, version 4 (MOZART-4), Geoscientific Model Development, 3(1), 43-67, https://doi.org/10.5194/gmd-3-42-2010, 2010.

Emmons, L. K., Schwantes, R. H., Orlando, J. J., Tyndall, G., Kinnison, D., Lamarque, J.-F., et al.: The Chemistry Mechanism in the Community Earth System Model version 2 (CESM2). Journal of Advances in Modeling Earth Systems, 12, e2019MS001882. https://doi.org/10.1029/2019MS001882, 2020.

Forster, P.M., Richardson, T., Maycock, A.C., Smith, C.J., Samset, B.H., Myhre, G., Andrews, T., Pincus, R., and Schulz, M.: Recommendations for diagnosing effective radiative forcing from climate models for CMIP6, Journal of Geophysical Research: Atmospheres, 121(20), 12460-12475, https://doi.org/10.1002/2016JD025320, 2016.

Squire, O.J., Archibald, A.T., Griffiths, P.T., Jenkin, M.E., Smith, D., and Pyle, J.A.: Influence of isoprene chemical mechanism on modelled changes in tropospheric ozone due to climate and land use over 21st century, Atmospheric Chemistry and Physics, 15, 5123-5143, https://doi.org/10.5194/acp-15-5123-2015, 2015.

von Kuhlmann, R., Lawerence, M.G., Pöschl, U., and Crutzen, P.J.: Sensitivities in global scale modeling of isoprene, Atmospheric Chemistry and Physics, 4, 1-17, 2004.

---

## Author Response (AR1)

Response to Reviewer #1

We thank the reviewer for their overall positive review and their constructive feedback. Below we provide point-by-point responses (in red) to the reviewer's comments (reproduced in black italics), with manuscript modifications indicated in **bold**.

*This manuscript presents original and valuable earth system model results based on sensitivity experiments to investigate the tropospheric VOC chemistry effect on climate. It is well structured, written and presented. I suggest acceptance of the manuscript for publication but I have a few minor comments to be considered.*

Thanks very much for the positive feedback, and we address your specific comments below.

*Comments*

*Something that I missed in the structure of the manuscript was a discussion of the induced radiative forcing due to the perturbation with explicit VOC chemistry (WACtl) versus the case with a simplified SOAG scheme and no explicit VOC chemistry (MACtl). Would you think that an estimate of effective radiative forcing from these two experiments would be feasible following the regression method of Gregory et al. (2004). For this method, it is the time development before the steady state is reached which is of interest and in your simulations the first 39 years of output (which were discarded because they showed significant trends in global mean surface temperature, indicating that the model had not fully equilibrated) may offer this opportunity.*

Thank you for this suggestion, as an analysis of the ERF would facilitate comparison with other climate forcers. Unfortunately, however, the regression approach of Gregory et al. (2004) is not well-suited for our model experiments. This is because there is initial warming of global mean surface temperature (GMST) over the first ~20 years of WACtl relative to MACtl (the reasons for which are unclear), even though the steady-state GMST response is negative. That is, the transient and equilibrium radiative forcings appear to be opposite in sign, and fixed SST simulations would be more suitable for calculating the ERF (e.g., Hansen et al., 2005; Forster et al., 2016), which is not a simple undertaking. We think that the complexity in the transient vs. equilibrium radiative forcing and the reasons for it are worth investigating in a separate study, and we have added text to this effect within the discussion of Fig. 1.

*line 130: I would suggest to add a few references for SOAG scheme implemented?*

Thank you for the suggestion. The following text has been added here: "**The simplified SOA scheme in MACtl is known as a SOAG scheme, and it is commonly implemented in GCMs. Under this scheme, anthropogenic and biogenic precursor VOCs are assumed to have fixed mass yields and are sorted into five primary VOC bins. Within MAM4, these gas-phase precursors are lumped into a single semi-volatile organic gas-phase species called SOAG, which then condenses into primary aerosol particles (Liu et al., 2012; Emmons et al., 2010).**"

*lines 198-199: The fact that the two cases were not initialized from the same ocean and sea ice states is a limitation. Although you mention that in the future, you plan to examine possible*

*sensitivity to different initialization approaches, could you speculate based on other studies on the degree that this could influence your results?*

Thank you for raising this point. The figure below shows the SST difference (WACtl – MACtl) averaged over just the first simulation day.

[Figure]

[Figure]

Figure R1-1: First day mean difference (WACtl – MACtl) in (a) SST (K) and (b) sea ice concentration (%).

The SST pattern is similar to the difference between a La Niña state and an El Niño state. Furthermore, the sea ice concentration differences appear to be mostly dictated by the

temperature differences, with lower (higher) sea ice concentration where temperature is warmer (colder). Overall, this analysis suggests that the difference between the WACtl and MACtl initial states is a difference in the phase of internal variability, rather than a fundamentally different climate state. Thus, MACtl and WACtl appear to originate from essentially indistinguishable climate states, even though their initial conditions are not precisely the same. Therefore, we expect that any long-term difference between MACtl and WACtl would be due to the effects of climate forcing during the simulations rather than climate differences in the initial states. If we were to perform another run, "WAmatched", with exactly the same initial conditions as MACtl, then we expect that the difference between WAmatched and WACtl would just be internal variability (similar to the difference between ensemble members in an initial condition ensemble), and the long-term average of WAmatched – MACtl would look essentially the same as WACtl – MACtl. Also note that the difference (WACtl – MACtl) between the long-term average of SSTs (Fig. 1 in the manuscript) does not resemble Figure R1-1a above, further indicating that the difference between WACtl and MACtl is not influenced by the difference in the initial states. We have added text to sections 2.4 and 3 to make these points.

*lines 297-299: The authors mention that " This makes sense because, if low cloud cover is fixed (which is approximately true around Antarctica) and SIC decreases (increases), then the radiative effect of low clouds gets stronger (weaker), implying a negative (positive) CRE change." Maybe the authors can clarify that this is related to albedo changes and not cloud fractions changes.*

Thank you for this point. We have modified the text here as follows: "This makes sense because, if low cloud cover is fixed (which is approximately true around Antarctica) and SIC decreases (increases), then **the surface albedo decreases (increases), and** the radiative effect of low clouds gets stronger (weaker), implying a negative (positive) CRE change."

*Figure 5: I am puzzling by the fact that the statistical significant near surface warming over the North polar regions seen in Figure 1 is not shown in Figure 5. Do you have any explanation for this inconsistency among the two figures?*

Below we show the same plot as in Fig. 5 but using a linear pressure scale, which reveals more detail in the lower troposphere.

[Figure]

Figure R1-2: Annual mean difference (WACtl-MACtl) in atmospheric temperature (K). The black contours indicate the MACtl climatology of air temperature (K), with a contour interval of 10 K. Gray crosses indicate regions that are statistically insignificant, with uncrossed regions being statistically significant at the 95% level. The thick black line marks the MACtl climatology of the tropopause. The pressure scale is linear.

This shows that there is indeed warming in the Arctic troposphere, but it is confined to very low altitudes, as we stated in the manuscript when discussing Fig. 5. So there is no inconsistency between Figs. 1 and 5.

*line 325: There is a weakening of the NH stratospheric polar jet but it is not clear the equatorward shift of the tropospheric mid-latitude jet.*

Thank you, we have modified the text here as follows: "In NH, there is a qualitatively similar weakening of the stratospheric polar vortex and **zonal wind deceleration on the poleward flank of the midlatitude jet. However, the zonal wind response is quantitatively weaker in NH than in SH, and there is no clear indication of an equatorward shift of the NH midlatitude jet.**"

*line 328: This is confusing as in the next sentence you clarify that the responses are opposite to the poleward shift due to global warming. Are you referring to stratosphere warming (as shown in Figure 5) or the thermal forcings that qualitatively mimic three key aspects of anthropogenic*

*climate change: (warming in the tropical troposphere, cooling in the polar stratosphere, and warming at the polar surface) discussed in Butler et al (2010) referenced here?*

Thank you for pointing out this possible point of confusion. The key thing to bear in mind from earlier studies is that the circulation responds differently depending on whether warming is confined to particular latitudes or spread over all latitudes. To avoid possible confusion, we have modified the text here as follows: "**Care is needed when comparing our zonal wind response to the responses in earlier studies, since the atmospheric circulation response is highly sensitive to the regional structure of temperature changes. For example, idealized simulations have shown that a narrow band of warming around the equator produces equatorward shifts of the midlatitude jets, whereas warming spread over all latitudes produces poleward shifts (Tandon et al., 2013). In our simulations,** the zonal wind responses in both hemispheres qualitatively resemble the circulation responses to high latitude warming found in earlier studies (e.g., Son et al., 2008; Butler et al., 2010), and we will explore the mechanisms generating this warming further below. Moreover, these responses are qualitatively opposite to the poleward midlatitude jet shifts produced in simulations of global warming **(with warming over all latitudes)**, including simulations in which SST is uniformly increased (e.g., Chen et al., 2013).

*lines 330-332: The authors mention that "based on such past work, we would expect that, aside from any regional temperature changes, widespread tropospheric cooling would also shift the midlatitude jets equatorward". Do you refer to past work related to the impact of aerosol cooling? Please prove some references.*

Thanks for this suggestion. We have added the following text here: "**Indeed, simulations of the response to increased natural aerosols have shown widespread tropospheric cooling and equatorward shifts of the midlatitude jets (Allen and Sherwood, 2011).**"

*line 342: The use of arrows on Fig.6b could help the reader to identify the counterclockwise anomaly extending poleward of the SH HC edge and a clockwise anomaly extending poleward of the NH HC edge. This is only a suggestion.*

Thank you for this suggestion. We considered adding arrows, but given the anomalies overlying the climatology, the figure appearance became too busy in our opinion. We hope that the text clarification in response to your next comment helps with interpreting the figure.

*lines 360-361: Pacific. The statement that "in addition to these HC changes, Fig. 6b also shows weakening of the Ferrel cells in both hemispheres and weakening of the polar cell in SH" needs more elaboration as it is not clear.*

We have clarified this point by adding the following text when discussing the HC changes: "**In general, weakening of a particular overturning cell is indicated by an anomaly over the central portion of an overturning cell that is opposite in sign to the climatological streamfunction. In this case, HC weakening in NH is indicated by a negative (counterclockwise) anomaly that opposes the positive (clockwise) HC climatology, and HC weakening in SH is indicated by a positive (clockwise) anomaly that opposes the negative**

**(counterclockwise) HC climatology."** Then when discussing the Ferrel and polar cell changes, the text has been updated as follows: "**As mentioned above, this weakening [of the Ferrel cells and SH polar cell] can be inferred from anomalies that are opposite in sign to the climatology.**"

*line 383: Please define the acronym SPS.*

Thank you for catching this. We have modified the text here as follows: "**Southern Polar Stratosphere (SPS).**"

*Line 405: The authors mention here that the warming in the Antarctica is likely explained by shortwave heating, which does increase. A link to Figure 8 that shows clearly this would be helpful for the reader.*

Thanks, we have added the following text here: "**(This increased shortwave heating is more clearly visible in Fig. 8, which plots the shortwave heating on a linear pressure scale with an adjusted shading scale.)**"

*line 448 : Are PANs included in NOy as part of organic nitrates?*

PANs are indeed included in the WACtl experiment as part of organic nitrates (Emmons et al., 2020). We have added text to lines 162-163 to clarify this.

*Line 547: The authors mention that in the midlatitudes of both hemispheres, the downward shifts of the clouds result in negative CRE as expected. This is clear for the SH but it is not clearly evident in the case for NH midlatitudes according to Figure 13.*

Thank you for pointing this out. We have modified the text here as follows: "In the **SH** midlatitudes **and NH high latitudes**, the downward shifts of the clouds result in negative CRE as expected."

References:

Allen, R.J., and Sherwood, S.C.: The impact of natural versus anthropogenic aerosols on atmospheric circulation in the Community Atmosphere Model, Climate Dynamics, 36, 1959-1978, https://doi.org/10.1007/s00382-010-0898-8, 2011.

Emmons, L.K., Walters, S., Hess, P.G., Lamarque J.-F., Pfister, G.G., Filmore, D., Granier, C., Guenther, A., Kinnison, D., Laepple, T, et al.: Description and evaluation of the Model for Ozone and Related chemical Tracers, version 4 (MOZART-4), Geoscientific Model Development, 3(1), 43-67, https://doi.org/10.5194/gmd-3-42-2010, 2010.

Emmons, L. K., Schwantes, R. H., Orlando, J. J., Tyndall, G., Kinnison, D., Lamarque, J.-F., et al.: The Chemistry Mechanism in the Community Earth System Model version 2 (CESM2). Journal of Advances in Modeling Earth Systems, 12, e2019MS001882. https://doi.org/10.1029/2019MS001882, 2020.

Forster, P.M., Richardson, T., Maycock, A.C., Smith, C.J., Samset, B.H., Myhre, G., Andrews, T., Pincus, R., and Schulz, M.: Recommendations for diagnosing effective radiative forcing from climate models for CMIP6, Journal of Geophysical Research: Atmospheres, 121(20), 12460-12475, https://doi.org/10.1002/2016JD025320, 2016.

Hansen, J., Sato, M., Ruedy, R., Nazarenko, L., Lacis, A., Schmidt, G.A., Russel, G., Aleinov, I., Bauer, M., Bell, N., et al.: Efficacy of climate forcings, Journal of Geophysical Research: Atmospheres, 110, D18104, 2005.

We sincerely thank Alexander for his valuable feedback that will help to improve our manuscript. Below we provide point-by-point responses (in red) to Alexander's comments (reproduced in black italics), with modifications to the manuscript indicated in **bold**.

*Stanton and Tandon present numerical model calculations, which aim to attribute the role of Volatile Organic Compounds (VOC) on climate. This is a worthwhile question and progress on the topic is needed inlight of efforts like the WCRPs Lighthouse Activity on "Explaining and Predicting Earth System Change". However, this study falls significantly far from being able to answer the question. I think the authors should be comended for their attempt with the modelling but significant flaws in the model design mean that they are unable to answer the problem they have set out to. I would suggest a significant revision, including a change of the title of the manuscript if the present results are to be used as the underpinning data for a revision.*

*Good points:*

*Overall the manuscript is well written. The climate analyses are excellent and the team are clearly experts in this area.*

Thank you for this positive feedback.

*Bad points:*

*The experiment design is very much flawed in being able to answer the question of VOCs impacts on climate. We have known for a long time that the effect of VOCs is non-linear on key climate forcers (e.g., O3, CH4, aerosols). By removing the VOCs one is significanlty perturbing the system into a state which does not allow the role of VOCs to be teased out. The study answers the question: "What are the impacts of removing VOC chemistry on pre-industrial climate simulated by CESM2". This is a useful question but far from the grander question raised by the title. I think this point is a critical one that can not be addressed without addressing the core of the study/revising the aims and scope.*

Thank you for this feedback. We agree that the influence of VOC chemistry would depend on levels of other atmospheric constituents like ozone, methane and aerosols. For this reason, we do not think it would be feasible in one paper to answer the general question of how VOC chemistry influences climate across all possible climate states. For the question to be well-posed, the base climate state needs to be specified. We make clear in the abstract and text that we focus only on preindustrial simulations, which is such a standard assumption for a base climate state that we didn't think it was necessary to specify that in the title as well. But prompted by your feedback, we have modified our title to be "How Does Tropospheric VOC Chemistry Affect Climate? An Investigation **of Pre-Industrial Control Simulations** Using the Community Earth System Model Version 2." This hopefully makes it absolutely clear that we are looking at just preindustrial climates.

As for your comment that "By removing the VOCs one is significanlty perturbing the system into a state which does not allow the role of VOCs to be teased out." We agree that removing

VOCs entirely would be a potentially extreme perturbation to the chemistry-climate system, with a whole chain of possible effects that would likely be hard to disentangle. But it would be misleading to say that VOCs are absent in our MACtl simulation. Rather, in MACtl, the explicit chemical reactions involving VOCs are absent, and the SOAG scheme attempts to capture the effects of VOCs (Liu et al., 2012; Emmons et al., 2010). So the difference between WACtl and MACtl is not the presence of VOCs but rather the presence of explicit VOC chemistry. We have added text to Section 2.2 (lines 141-144) to make this clear.

*As someone who has looked at aspects of the title problem I am very surprised by the large changes in surface temperature that the authors show in Figure 1. The small difference in Global Mean Surface Temperature (GMST) suggests that the surface response is really not that signficiant. How does that compare to the spread of PI GMST simulated by CMIP6 models or even ensemble members of CESM2? Are these changes really significantly different from the uncertainty in the literature? The model we use in the group, UKESM-1-LL, shows significant variance across ensemble members (even under PI conditions). As it currently stands, I don't think the results suggest that there is a significant imapct of VOC chemistry on climate; BUT as I said I am not sure that the experiment design allows one to answer this question.*

Thank you for this feedback. Hopefully, our response to your previous comment clarifies our experimental setup's relevance to the scientific question. Given that SOA effects can be highly regional, we should not necessarily expect a large GMST response to VOC chemistry. While our statistical significance tests provide some confidence that our surface temperature responses are significant, we can show some additional results to increase confidence in the robustness of our results. First, we have extended our simulations by 70 years so that the averaging period is now 211 years. Our revised manuscript will show results from these extended simulations, and the updated Fig. 1 is reproduced below as Fig. R2-1. All of the key features in these results are the same as in the submitted manuscript, indicating that internal variability was not strongly influencing our results.

[Figure]

Figure R2-1: Annual mean difference (WACtl – MACtl) in surface temperature (K) from 211 years of model output. Gray crosses indicate responses that are not statistically significant, with uncrossed responses being statistically significant at the 95% level.

We do not think it would be particularly helpful to look at the spread among different CMIP6 models, as that spread can be due to intermodel differences in climate states, not just internal climate variability. Furthermore, in the CESM2 large ensemble, the atmospheric component was the Community Atmospheric Model (CAM), not WACCM, so it would not cleanly compare with our simulations. The CMIP6 archive does include a preindustrial control simulation using the same configuration as our WACtl simulation, but it is only 250 years long, which is not long enough to be helpful for this discussion. However, the CMIP6 archive also includes a 500-year preindustrial control simulation using almost the same configuration as our MACtl simulation, except that it includes an interactive QBO, and we call this simulation "MACtl-CMIP6." (Note that in our initial response to the reviewer, we mistakenly stated that this simulation uses the same configuration as MACtl. Nonetheless, we have found that the QBO treatment makes no difference to the surface temperature response, and we have added text to this effect at lines 265-267.) This MACtl-CMIP6 simulation is long enough to give an indication of internal variability when averaging over time periods comparable to the lengths of our simulations.

Below is the difference in surface temperature when averaging over non-overlapping 211-year time slices in this CMIP6 control run of CESM2-WACCM6. Each of the panels shows a different choice of averaging period, and they are plotted on the same shading scale as used in Fig. 1 of our manuscript and Fig. R2-1 above. None of these calculations shows temperature differences comparable to the significant temperature differences shown in Fig. 1. This analysis further establishes that the responses shown in our simulations are not due to internal climate variability, and the responses represent the preindustrial climate effects of VOC chemistry. We have added this analysis to section 3 of the manuscript, and Fig. R2-2 below is included in the manuscript as Fig. 2.

[Figure]

Figure R2-2:

Difference of time-averaged surface temperature between non-overlapping 211-year time slices of the CMIP6 preindustrial control simulation of CESM2-WACCM6-FV2 (variant label r1i1p1f1). This simulation uses the same configuration as our MACtl simulation. The precise simulation years that were averaged are indicated in the title over each panel.

*What was also lacking for me was more of a focus on the causal links between tropospheric composition changes and their impacts on radiative forcers. There is clearly a very large response in OH. Why? I would guess the removal of isoprene, which has been addressed before, many times (see e.g., Bates and Jacob, 2019; Squire et al., 2015; von Kuhlmann et al., 2004). I was surprised not to see mention of isoprene at all in any analyses. Moving from OH one can then identify the impacts of changes in oxidising capacity on aerosols and aerosol precursors. What happens to SO2? Why does Sulfate change the way it does? Some of this can only be understood by constructing budgets of the variables (production and loss) and analysing them. Like I said the climate analysis is good but the attribution to composition changes leaves a lot to be desired and can be thoroughly improved to provide insight into causality (but not attribution of it!).*

Thank you for this feedback. Just to clarify, isoprene chemistry is included in WACtl but not in MACtl. Yes, the addition of isoprene chemistry in WACtl likely explains the large decrease in OH relative to MACtl. We have added details of the impacts of isoprene chemistry in the results section under figure 10 along with references to the earlier studies you mention (Bates and Jacob, 2019; Squire et al., 2015; von Kuhlman et al., 2004). For example, at lines 541-543 the text now reads as follows: "**In particular, the reduction of tropospheric OH concentration due to the inclusion of isoprene chemistry has been well documented in the literature (e.g., von Kuhlman et al., 2004; Squire et al., 2015; Bates and Jacob, 2019).**" The influence of OH versus precipitation on $SO_2$ and sulfate changes will be addressed in a response to a comment below.

*In additio to these minor comments I have more minor comments:*

*L26: The abstract is missing a conclusive statement at the end.*

The following statement has been added to the end of the abstract: "**Some of the climate responses are quantitatively large enough in some regions to motivate future investigations of VOC chemistry's possible influences on anthropogenic climate change.**"

*L38: I don't think oxidization is a word. Change to oxidation.*

Thank you for catching this, we have corrected this to "**oxidation**.**"**

*L41: Not clear how RO2 makes NO3. Add a reaction or reference. Need to define HO2 and RO2.*

Sorry for the confusion here. We should have said $RONO_2$ instead of $NO_3$ here. The manuscript has been modified here as follows: "These peroxy radicals **($RO_2$)** can participate in further oxidation reactions, producing a range of products, such as **organic nitrates ($RONO_2$) and hydroperoxyl radicals ($HO_2$)**."

*L45: Not only "typically". Actually. That's the nature of a CTM. See Young et al (2018) for an overview of the types of models and feedbacks/couplings and adopt that nomenclature.*

Thank you, we have changed "is typically" to "**by design is**" and we have inserted the CTM acronym here in order to adopt standard nomenclature.

*L56: They didn't have to prescribe SSTs. They chose to. there are still elements of climate response with fixed SSTs and the use of fixed SSTs is the defacto method for calculating key climate metrics like ERF.*

Thank you, we agree. The text here has been changed to the following: "Climate simulations with more comprehensive chemistry **typically prescribe sea surface temperatures (SSTs) (e.g., Tilmes et al., 2019), which facilitates calculating standard climate metrics like effective radiative forcing (ERF) (e.g., Forster et al., 2016)**, **but this approach** limits the ability…".

*L61: Define NOx.*

On L61 the following has been added after $NO_x$: "**(NO + NO₂)**".

*L73: …limits the model's ability to produce a "full" climate response. Add "full".*

Thanks, we have added "**full**" here.

*L84-85: Absolute temperature variations across climate models is way larger than this. Anyway, what is key to climate change is the difference between simulations under different forcings within a model. This statement can be misconstrued and so needs to be toned down, alot. I still don't see the surface temperature response as being at all significant so would like to be convinced more on this point.*

Our response to your earlier comment hopefully provides reassurance about the robustness of our results within the framework of CESM2-WACCM6. (And we agree, the spread across CMIP6 models isn't the key consideration here.) We see that our statement here could be misconstrued to challenge the role of greenhouse gases in future climate change. So, we have modified the text here as follows: "When the effects of tropospheric VOC chemistry are isolated, our simulations show significant impacts on climate, comparable in some regions to the temperature changes **expected from anthropogenic greenhouse gas increases** over a century in future climate change scenarios."

*L87: Typically RONO2 are formed from RO2+NO too. Should be made clear if BVOC+NO3 is the only route to RONO2 or if there are other routes, too. And what Organic Nitartes are comprised of in the model. PAN?*

$BVOC + NO_3$ is not the only route for $RONO_2$ in the WACtl case. $RO_2 + NO$ is also included. PAN is included in WACtl, but not in MACtl. In WACtl, PAN is part of organic nitrate (Emmons et al., 2020), and we have updated the manuscript at lines 162-163 to clarify this.

Based on further investigation of how NOy is treated in the model, we have updated much of this section. For example, we recently learned that, while gas phase NOy is included in the model, the condensed phase (which impacts radiation) is lumped with sulfate aerosols. This matter will also be addressed in a comment below.

*Table 1: It's unclear how key reactions, like the CH4 oxidation, varies between experiments. Neither is it clear what the emissions of NOx and VOCs are.*

The CH$_4$ oxidation reactions are the same in WACtl and MACtl, with the only difference being a CH$_4$ production term in the WACtl case. This production term enters into the reaction of propene with ozone (O$_3$ + C$_3$H$_6$ → C$_2$H$_4$O$_2$ + HCHO, C$_2$H$_4$O$_2$ → CH$_4$ + CO$_2$). We have updated the text at lines 135-136 to clarify this.

The emissions of VOCs and NOx are addressed in section 2.3, and we have added text to lines 205 and 212 to make this clear.

*L143: So there is no methane? No CO? Why? NB I am not saying CO is a VOC.*

CH$_4$ and CO are solution species in both MACtl and WACtl. The manuscript has been updated here to make this point about methane clear (line 134): "without any explicit **non-methane VOC** reactions."

*L175: I agree HONO is important but many things are missing that are important for ozone (see for example my review paper on tropospheric ozone, Archibald et al. 2020). The focus on HONO seems quite parochial.*

Thank you for this comment. A large majority of the species present in Table 1 of Archibald et al. (2020) are present in WACtl except HONO, CH$_3$CH(OO)CH$_3$, CH$_3$CH(OOH)CH$_3$, C$_2$H$_5$CO, C$_2$H$_5$C(O)OO, PPAN, MeONO$_2$ (assuming this is CH$_3$ONO$_2$), MSA, and DMSO. To our knowledge, besides HONO, none of these additional species are expected to be important for ozone in the absence of recent anthropogenic emissions. Nonetheless, we have added text to the conclusions (lines 683-686) to acknowledge that some compounds absent from CESM2-WACCM6 (e.g., methyl nitrate and PPN) should be considered in simulations that include recent anthropogenic emissions (which our simulations exclude). Note that in Table 1 of Archibald et al. (2020) there may be a typo. The variable named MGLY has the formula (CH$_3$COCHHO), but it appears this is meant to be methyl glyoxal, which has the formula (CH$_3$COCHO).

*L198: I think this is a major weakness of the study. I would welcome comments on the limitiations that this imposes and whether 39 years is really long enough for a fully coupled model to "spin-up".*

Neither of our model simulations was "spun-up" from a rest state. Rather the WACtl case was initialized from a climatological ocean and sea ice state, and the MACtl run was initialized from a reference case of the same model configuration that was already spun up. As we showed in our response to the Anonymous Reviewer (see Fig. R1-1), the difference between the initial SST states appears to be a shift in the phase of internal variability (specifically El Niño-Southern Oscillation, ENSO), rather than a fundamental difference in the long-term climate state. So 39 years is a reasonable time period to allow for WACtl to adjust from its initial state, as that adjustment would mostly involve an adjustment to VOC chemistry. We have updated the manuscript in section 2.4 to give more discussion of the initial states and their differences.

*Figure 1: This is rather puzzling a result. For a start the largest changes are over the oceans. This is where some fixed SST runs would help to remove any noise caused by changes in ocean circulation which will have a long time signal (and require more than 140 years of run). The main changes in temperature seem associated with sea ice zones. Is that correct? What could*

*cause such large and significant ~ grid-box level changes in surface temperature over land? I can't think of a mechanism associated with chemistry.*

We thank you for this feedback. While running fixed SST simulations would reduce noise, it would greatly limit the full climate response, which is what we are interested in. Hopefully our response to your earlier comment addresses any concern that our responses are strongly influenced by internal variability. While the most significant surface temperature changes are in the extratropics, these changes are not confined to sea ice zones.

We are not sure of the mechanisms driving the localized surface temperature changes over land. Most of our investigation of mechanisms was focused on the zonal mean changes, and we plan to investigate more localized changes in future studies. We have updated the text at lines 250-251 to make this point.

*Figure 1-Figure 6 and results and discussion. How large are these changes compared to for example the spread of CMIP6 models and or the spread of LENS/2 simulations with CESM/2.*

Please refer to our earlier response and Figure R2-1 regarding the comparison of the surface temperature response to internal variability. As we stated before, comparisons to the spread of CMIP6 models and LENS simulations would not be conclusive here, but we provided comparison to a longer CMIP6 control simulation using CESM2-WACCM6.

*L445: Are RONO2 coupled to the radiation scheme?*

Thank you for this question. As we stated above, we investigated further and found out (through correspondence with the modelling centre) that gas phase $RONO_2$ is not coupled to the radiation scheme, and condensed phase $RONO_2$ is lumped with sulfate aerosol (which does impact radiation). So the radiative effects of condensed organic nitrates are not treated separately from those of sulfate aerosols. This updated understanding required a number of updates to our manuscript, as we can now be more confident that the widespread cooling in the extratropical troposphere is likely due to increased sulfate aerosols in the stratosphere.

*Figure 9-12: % changes would be much more helpful. Please also express species in units that are more widley used in the literature. pptw for example is not used in atmospheric chemistry circles. Instead use mass per unit volume (g/m3 for example).*

Thank you for these suggestions. We have updated these figures (now Figures 10-13) to show percent changes and use more standard units for the climatology. For gas-phase species (e.g. OH in Fig. 10b), the usage of pptv still seems appropriate.

*L465: What matters more is how SOA are dealt with in the model, not what the literature says. Can you please expand on the coupling of SOA to radiation (through direct and indirect effects).*

We appreciate this feedback. SOA is indeed coupled to the radiation scheme (RRTMG) in CESM2-WACCM6 in both the WACtl and MACtl cases. This coupling is through both the direct effect (influencing aerosol optical depth) and the indirect effect (SOA acting as cloud condensation nuclei). Clouds are also coupled to RRTMG. This has been clarified in the methods

and results sections. We have updated the text in both the methods and results sections to make these points.

*L483:  This causality is not possible to determine. One would need to isolate ONLY the VBS to assert this.*

Thank you for pointing this out. The language of causality will be removed here. The text here has been updated as follows: "The VBS mechanism for SOA formation **would be expected to act as a sink for various radical species (OH, NO$_3$, NO, NO$_2$ and Cl). (To establish this effect with greater certainty, we would need to perform additional simulations that isolate the VBS mechanism, which is left for future work**.)"

*L523: This can be because: 1) you have less OH so less SO2 forms sulfate 2) you have more clouds and rain so more SO4 is wet deposited. Which is it? I think you need to examin the SO4 budget.*

We greatly appreciate these points as they have helped to strengthen our analysis. It appears that OH changes have a stronger influence compared to wet deposition changes. To provide evidence for this, a new figure has been added to the results section (Fig. 13), reproduced below, with accompanying text added to the manuscript.

[Figure]

Figure R2-3: (a) Annual mean difference (WACtl – MACtl) in vertical column density (VCD) of sulfate aerosol (kg m$^{-2}$), (b) VCD of SO$_2$ (kg m$^{-2}$), (c) zonal mean precipitation (mm d$^{-1}$), and (d) VCD of OH (kg m$^{-2}$). The differences that are statistically significant at the 95% level are colored red.

Panel (a) indicates a decrease in sulfate aerosol over the tropics, where OH decreases, but there is no widespread increase in precipitation in the tropics. This suggests that the tropical decreases in sulfate are driven primarily by changes in OH rather than changes in wet deposition of SO$_4$. It is notable, however, that over approximately 2-25°N, there is a precipitation increase that aligns with a local minimum in the sulfate aerosol change. This suggests that increased wet deposition of SO$_4$ is also contributing to the sulfate aerosol decrease in particular regions. Also note, as discussed in the manuscript, that increased SOA is likely also contributing to these tropical sulfate decreases.

References:

Bates, K.H, and Jacob, D.J.: A new model mechanism for atmospheric oxidation of isoprene: global effects on oxidants, nitrogen oxides, organic products, and secondary organic aerosol, Atmospheric Chemistry and Physics, 19, 9613-9640, https://doi.org/10.5194/acp-19-9613-2019, 2019.

Emmons, L.K., Walters, S., Hess, P.G., Lamarque J.-F., Pfister, G.G., Filmore, D., Granier, C., Guenther, A., Kinnison, D., Laepple, T, et al.: Description and evaluation of the Model for Ozone and Related chemical Tracers, version 4 (MOZART-4), Geoscientific Model Development, 3(1), 43-67, https://doi.org/10.5194/gmd-3-42-2010, 2010.

Emmons, L. K., Schwantes, R. H., Orlando, J. J., Tyndall, G., Kinnison, D., Lamarque, J.-F., et al.: The Chemistry Mechanism in the Community Earth System Model version 2 (CESM2). Journal of Advances in Modeling Earth Systems, 12, e2019MS001882. https://doi.org/10.1029/2019MS001882, 2020.

Forster, P.M., Richardson, T., Maycock, A.C., Smith, C.J., Samset, B.H., Myhre, G., Andrews, T., Pincus, R., and Schulz, M.: Recommendations for diagnosing effective radiative forcing from climate models for CMIP6, Journal of Geophysical Research: Atmospheres, 121(20), 12460-12475, https://doi.org/10.1002/2016JD025320, 2016.

Squire, O.J., Archibald, A.T., Griffiths, P.T., Jenkin, M.E., Smith, D., and Pyle, J.A.: Influence of isoprene chemical mechanism on modelled changes in tropospheric ozone due to climate and land use over 21$^{st}$ century, Atmospheric Chemistry and Physics, 15, 5123-5143, https://doi.org/10.5194/acp-15-5123-2015, 2015.

von Kuhlmann, R., Lawerence, M.G., Pöschl, U., and Crutzen, P.J.: Sensitivities in global scale modeling of isoprene, Atmospheric Chemistry and Physics, 4, 1-17, 2004.

---

## Author Response (AR2)

We appreciate the additional feedback from Alexander Archibald. Below we provide point-by-point responses (in red) to the reviewer's comments (reproduced in black italics), with manuscript modifications in **bold**.

*Further points to make clearer/elaborate on in the conclusions:*

*1) Figure 1. I am still puzzled as to how there are such large and significant tas changes in small regions over land. I mean here the changes over South America, Africa, North America and China. The authors have stated "the localized mechanisms behind these changes are not clear and will be investigated in future studies" and I would really urge that. I just don't get how such a significant response can be generated from VOC chemistry and I think it would be useful if the authors expand in the conclusions in how they could determine the localized mechanisms and or how important the results of Figure 1 are to the topic under study?*

Thank you, we agree that this topic is deserving of further study, and we have added the following text to the Conclusions to emphasize this (lines 659-664):

"**While we explored the mechanisms of the large-scale temperature changes, our simulations also show significant localized temperature changes over land. The reasons for these localized changes are unclear, and this is an important topic for follow-up study. Such investigation would require analysis of regional changes in shortwave, longwave and dynamical heating, going beyond our zonally averaged analysis of these quantities.**"

*2) Experiment design. The authors have tried to focus on VOC effects on climate and I really do love this. But this is a can of worms in my mind. Figure A2 underscores that there are significant ozone differences that are present in the models. If we want to attribute the role of VOCs would we not be better if we could do that by fixing the ozone too? Otherwise are we not looking at the climate response to different ozone fields? How much of the spread in ozone is due to the VOC chemistry and how much is due to other differences in the model set up? Again,I think a further comment on how experiments could be designed to better investigate these issues in the future would be very valuable.*

Thank you for this feedback. Our analysis showed evidence that ozone changes likely do not drive the climate response, as the strong warming in the Southern Polar Stratosphere (SPS) is mostly driven by dynamical heating. Nonetheless, in future studies, it is worth doing controlled experiments with prescribed ozone to confirm that the ozone changes do not have a strong climate effect. Furthermore, the ozone changes are interesting in their own right and deserving of further study. We have added text to this effect in the Conclusions (lines 681-685).